# On Occlusions in Video Action Detection: Benchmark Datasets And Training Recipes

**Rajat Modi**[1]*, **Vibhav Vineet**[2], **Yogesh Singh Rawat**[1]
CRCV, University of Central Florida[1], and Microsoft Research[2]

## Abstract

This paper explores the impact of occlusions in video action detection. We facilitate this study by introducing five new benchmark datasets namely O-UCF and O-JHMDB consisting of *synthetically controlled* static/dynamic occlusions, OVIS-UCF and OVIS-JHMDB consisting of occlusions with *realistic motions* and Real-OUCF for occlusions in *realistic-world* scenarios. We formally confirm an intuitive expectation: existing models *suffer a lot* as occlusion severity is increased and exhibit *different behaviours* when occluders are static vs when they are moving. We discover *several intriguing phenomenon* emerging in neural nets: 1) transformers can *naturally outperform CNN models* which might have even used occlusion as a form of data augmentation during training 2) incorporating symbolic-components like capsules to such backbones allows them to *bind* to occluders *never even seen* during training and 3) Islands of agreement *can emerge* in *realistic* images/videos *without* instance-level supervision, distillation or contrastive-based objectives[2](eg. video-textual training). Such emergent properties allow us to derive simple yet effective training recipes which lead to robust occlusion models inductively satisfying the *first two stages* of the binding mechanism (grouping/segregation). Models leveraging these recipes *outperform* existing video action-detectors under occlusion by 32.3% on O-UCF, 32.7% on O-JHMDB & 2.6% on Real-OUCF in terms of the vMAP metric. The code for this work has been released at `https://github.com/rajatmodi62/OccludedActionBenchmark`.

## 1 Introduction

Deep learning[41] has led to significant advances in object detection/segmentation for both image[16, 19] and video domain[84]. Such deep neural networks are in turn widely used in self-driving cars and safety critical scenarios. A key concern for such applications is whether they are able to perform well when encountering realistic occlusions: e.g. are they able to reliably localize a pedestrian even when an occluder (say a dog) comes in front of him. However, one major limitations being existing dataset test split doesn't contain such occlusions. This raises a concern, whether these models will be robust to real-world occlusions or not.

One would suspect that the inherent inductive biases of these architectures would be enough to induce natural occlusion robustness. To verify the hypothesis, we run two *preliminary setups:* (Fig. 1) (A) Firstly, superimposing a single occluder (eg bus) over the actor, and, (B) Then, we analyze the performance if the occlusion is shifted to background (i.e. no occlusion over actor at all). We observe a relative drop of 20-50% across multiple existing state-of-the-art approaches [46, 13]. This verifies

---

*Corresponding Author, email: rajatmodi@ucf.edu

[2]Grouping pixels is a perceptual phenomenon of the visual-cortex[59] . Assigning a group to a class is a property of language. If a group is identified by multiple-names, then a neural-net can learn this one-many mapping via contrastive objectives[75]. But grouping should *also* emerge *without* language since organisms can continue to perceive objects even without ears/tongue.

37th Conference on Neural Information Processing Systems (NeurIPS 2023) Track on Datasets and Benchmarks.

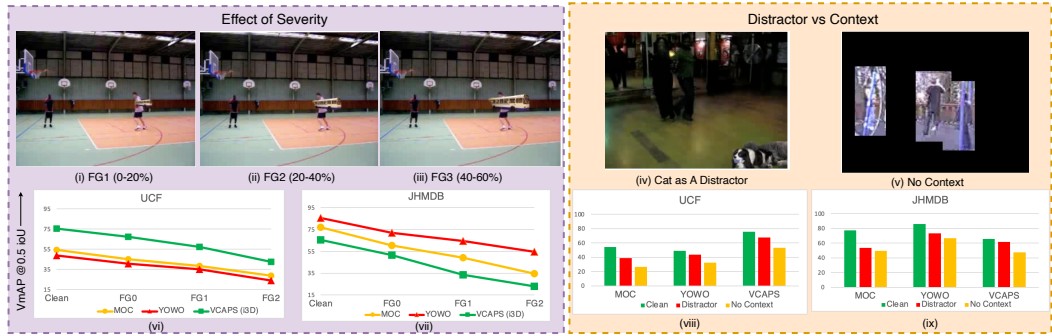

Figure 1: **A Toy Experiment.** (i-iii) superimposing a single occluder (bus) on an actor and varying its size results in drops as large as 50%. (iv) even a simple occluder (cat) in background results in 30% drop. (v) highest drops are observed if background is entirely masked. (vi-ix) Clean refers to all methods evaluated on unoccluded test sets. Best viewed when zoomed in.

our hypothesis that existing approaches are *not* robust to occlusion. It opens up the area that there is a need to study these models behaviour under different types and severity of occlusion.

We conduct *first* benchmark study of occlusions in spatio-temporal video action detection. We investigate following aspects: a) *Effect of different severity level of occlusions*, b) *Robustness of different network backbones*, c) *Effect of pre-training and trainable parameters*. d) *Effect of static vs dynamic occlusions on models*. Our study yields the following key insights: 1) Performance decreases when severity increases on actor region, but increasing severity on background does not have much impact 2) Transformers which have *never* seen occlusions during training can outperform CNN-based models which *uses* occlusions as a data augmentation. 3) a significant component of transformer's robustness (and other architectures as well) comes from pre-training on large-scale datasets[7]. 4) Existing models perform better on static occlusions than *realistic* dynamic occlusions.

Based on our study, we investigate techniques to make the existing video action detection models robust against occlusion. First, we experiment with augmentations and found them to be effective for all the models in improving their robustness. We further utilize some of these insights and develop a transformer-based token masking strategy that provides a robust video action detection model which outperforms all existing methods. We make the following contributions in this work:

- We analyze existing approach behaviours on occlusions and conduct a *first* benchmark study on occlusions in spatio-temporal action detection.

- Towards this, we propose *five* novel datasets, i.e. O-UCF, O-JHMDB for static occlusions, OVIS-UCF, OVIS-JHMDB for *realistic* occluder motions and Real-OUCF for real world occlusions to benchmark our study.

- To improve robustness of existing approaches, we show how occlusion as augmentation improves robustness. Furthermore, we propose token-masking to further improve robustness of transformer backbones.

## 2   Related Work

**Spatio-Temporal Action Detection (STVAD):** Involves localizing an actor in *each* video frame and also predicting an action class . Existing methods[66, 58, 11] adopt a tubelet- based approach: action trajectories can be thought of as a sequence of actor bounding boxes spread over time. These tubes can be predicted by progressively refining predictions over a center-keyframe[46, 77] or directly learning a tubelet as a feature volume[86]. However, these methods require additional post-processing by linking these smaller tubelets across time. Other architectures like VideoCapsuleNet [13] frame action-detection as a per-pixel localization task.

**Occlusions in Videos:** Recently, OVIS dataset[54] was proposed for Video Instance Segmentation (VIS) under realistic occlusions. Methods like [76, 32, 3] introduce occlusion robustness by temporally averaging actor-centric embeddings in DETR-like architectures[5]. However, instance-recognition during VIS can be performed by object-detection in a single frame whereas action-classification in STVAD requires more than one frame. Approaches like [44, 70] explore occlusions

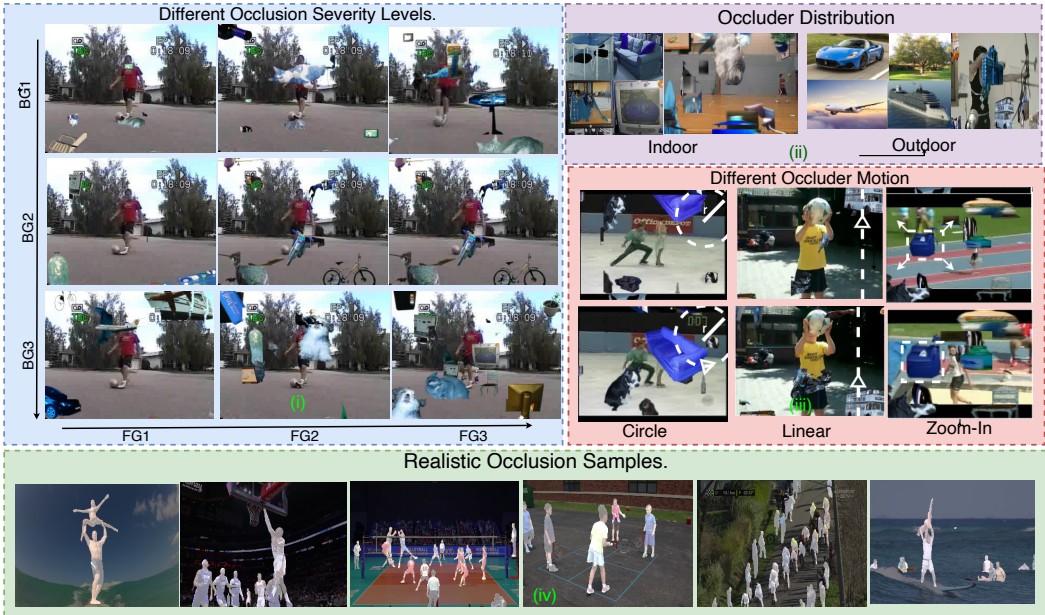

Figure 2: **Sample video frames from proposed benchmark datasets.** (i) occlusion severity increases across 9 severity levels and both actor/background region. (ii) occluders are sampled from both indoor/outdoor splits. (iii) Our O-UCF and O-JHMDB simulate 6 dynamic occluder motions like circle, linear, zoom-in etc. (iv) Similary, the proposed Real-OUCF is a dataset for realistic scenarios where multiple actors mutually-occlude each other. Best viewed in color.

for action-recognition. In comparison, we *additionally* focus on the impact of occlusions on *localization* ability of SOTA action-detectors.

**Occlusion in Images:** CompositionalNets[40] explore occlusion-robustness for image classification through EM-based mixture models. DeepVoting [85] caters to object-detection under occlusions by proposing a voting layer to gather contributions from visible object parts. While such studies are restricted to images, we study occlusions under videos.

**Regularization Tenchniques:** Dropout [62] have been shown to act as an implicit regularizer [69] and reduce network overfitting. Moreover, dropout has been found effective for increasing occlusion robustness [21]. We *further* show that masking input tokens in transformers [51] is also an effective training strategy for this task.

## 3 The Video Occlusion Datasets

The aim of this study is to study occlusions in spatio-temporal video action detection. We present five different benchmark datasets to study this problem. Two of them (O-UCF and O-JHMDB) are synthetically generated to systematically study this problem by manipulating a single occlusion parameter at a time. Similarly, OVIS-UCF, OVIS-JHMDB consist of occluders exhibiting realistic motions. Furthermore, we also present a real-world dataset (Real-OUCF) to validate our findings. We use three different parameters to create our synthetic datasets, 1) Indoor/Outdoor Occluders 2) Severity of Occlusion 3) Static/Dynamic motions of occluders.

### 3.1 Occluder Selection

To keep the generated occlusions *realistic*, we first crop out several objects from the Pascal-VOC[14] images using ground-truth pixel-level annotations. Out of 2413 cropped instances, we shortlist 900 possible occluders which correspond to commonly-occurring objects. Next, these objects are distributed among two separate categories namely indoor objects ( eg, chair, table) and outdoor objects (eg, aeroplane, ship etc.). Finally, we *softly* blend these occluders over the video pixels using RGBD maps of these objects. This blending correctly simulates a very real object (occluder) being present over in the frame, without having to resort to an intensive data collection procedure.

Table 1: Benchmark dataset statistics

| Statistics | O-UCF | OVIS-UCF | O-JHMDB | OVIS-JHMDB | Real-OUCF |
|---|---|---|---|---|---|
| Classes | 24 | 24 | 21 | 21 | 24 |
| Severity Levels | 9 | 9 | 9 | 9 | - |
| Occ. Motions | 6 | 6 | 6 | 6 | - |
| Occ. Type | Synthetic | Semi-Real | Synthetic | Semi-Real | Real |
| Total Videos | 20306 | 20306 | 5896 | 5896 | 1743 |
| Instances | 2284 | 2284 | 928 | 928 | 6920 |

## 3.2 Severity of Occlusion

In a video, an actor occupies certain portion of the frame as it moves over time. Formally, we define such an actor-region (FG) as the tightest bounding box enclosing all the ground truth boxes of the actor's trajectory over time. The remaining region is termed as background(BG).

Occlusion severity is defined as the fraction of the total area which is occupied by an occluder during occlusion. We experiment with three severity levels, namely 1 (0-20%), 2 (20-40%), and 3 (40-60%) respectively. Occlusion levels in the actor region are prefixed by FG, i.e. FG1/2/3, whereas occlusions in the background are prefixed by BG, i.e. BG 1/2/3. A combination of these actor/background severity levels yields 9 levels of severity.

## 3.3 Types of Occlusion

**Static Occlusions:** Refer to the occlusions where the occluders occupy a fixed position in the frame and don't move over time. Our dataset consists of 9 severity combinations of actor/background regions as explained in the previous section.

**Dynamic Occlusions:** Existing occlusion-based datasets are mostly image-based and are therefore unable to study the effects of the movement of occluders in the video. For such dynamic cases, we pick a fixed severity level (i.e. FG2, BG3) and vary the motion of the occluders from a particular starting point. Our test set consists of two types of motions, i.e. circular and sinusoidal. The training set consists of linear, zoom-in, zoom-out, or random motions. Note that the motion in the train/test set is mutually-exclusive, i.e. one type of motion in one dataset split is not present in another for fairness. Similarly, an occluder moving in the actor region never mixes over to the background region but instead gets wrapped around over the course of its trajectory.

## 3.4 Benchmark datasets

The statistics for five proposed benchmark datasets are illustrated in Table 1. **O-UCF** contains of 24 action classes, along with 20306 testing samples. Similarly, **O-JHMDB** contains 21 action clases with 928 samples. Both of these datasets consist of static occlusions in 9 levels of severity, 4 *controlled* occluder trajectories in train set, and 2 trajectories in the test set. Our **OVIS-UCF/OVIS-JHMDB** datasets consist of occluders with realistic motions from OVIS superimposed on top of original UCF/JHMDB datasets. Note that we term this dataset as semi-realistic because although the occluders themselves are synthetically placed on the frames, occluder trajectories are naturally occuring as observed in OVIS dataset. **Real-OUCF** consists of 1743 fully-realistic occlusion videos which were hand-picked from Youtube for 24 action classes. These videos were then cropped temporally using LossLessCut[2], to precisely localize start/end time step of each action. Such shorter duration clips then need to be spatio-temporally annotated. Therefore, we feed-forward all such clips through auto-label generator of GroundedSAM[48] in order to localize "person" class with an appropriately constructed textual query. Since SAM[38] is a foundational model, it segments all the persons in an image. However, we are only concerned with the people who are actually performing actions. In order to remove such excessive false positives predicted by GroundedSAM[48], we manually suppress/refine per-frame instance-level masks using the CVAT Annotation Tool[1]. Finally, we end up with 64.1% of annotated instances being occluded.

# 4 Experiments and Analysis

**Studied models:** We experiment on three most recent SOTA methods in video action detection. Namely, we pick MOC[46],YOWO[39] and VideoCapsuleNet[13] whose official Github implementations and evaluation protocols have been *fully* open-sourced for *both* UCF and JHMDB datasets. All

Table 2: **O-UCF Benchmark:** Performance across three occlusion severity levels. For a particular actor-region occlusion (FG), averaged results across BG1/2/3 levels are reported.

| | Clean | FG1 | | | FG2 | | | FG3 | | |
| --- | --- | --- | --- | --- | --- | --- | --- | --- | --- | --- |
| | | Occ | $\delta_a$ | $\delta_r$ | Occ | $\delta_a$ | $\delta_r$ | Occ | $\delta_a$ | $\delta_r$ |
| MOC[46] | 54.4 | 53.6 | 0.99 | 0.99 | 35.2 | 0.81 | 0.65 | 29.5 | 0.75 | 0.54 |
| YOWO[39] | 48.8 | 38.5 | 0.90 | 0.79 | 32.5 | 0.84 | 0.67 | 26.8 | 0.78 | 0.55 |
| VCAPS[13] | 75.5 | 51.5 | 0.76 | 0.68 | 42.8 | 0.67 | 0.57 | 36.9 | 0.61 | 0.49 |

Table 3: **O-JHMDB Benchmark:** Performance across three occlusion severity levels. For a particular actor-region occlusion (FG), averaged results across BG1/2/3 levels are reported.

| | Clean | FG1 | | | FG2 | | | FG3 | | |
| --- | --- | --- | --- | --- | --- | --- | --- | --- | --- | --- |
| | | Occ | $\delta_a$ | $\delta_r$ | Occ | $\delta_a$ | $\delta_r$ | Occ | $\delta_a$ | $\delta_r$ |
| MOC[46] | 77.2 | 45.2 | 0.68 | 0.59 | 33.9 | 0.57 | 0.44 | 25.9 | 0.49 | 0.34 |
| YOWO[39] | 85.7 | 50.5 | 0.65 | 0.59 | 47.7 | 0.62 | 0.56 | 46.2 | 0.61 | 0.54 |
| VCAPS[13] | 65.7 | 49.2 | 0.84 | 0.75 | 35.4 | 0.70 | 0.54 | 21.7 | 0.56 | 0.33 |

these models are tested on the original *clean* UCF24 & JHMDB-21 datasets, which do not contain any occlusions. Also, we evaluate these models on *occ* versions, i.e. our O-UCF/O-JHMDB datasets. All our transformer-based models have been trained for 1.5x more epochs than CNN based models to facilitate appropriate convergence. The best performing models for downstream testing are selected from a separate hold-out validation set.

**Metrics** A well accepted metric for action-detection is v-mAP at 0.5 ioU threshold, as it also measures the *temporal consistency* of the prediction across time. Assume that under occlusions vmAP drops from $V$ to $V'$. Therefore, we report *absolute* robustness $\delta_a = (1 - \frac{V-V'}{100})$ and *relative* robustness $\delta_r = (1 - \frac{V-V'}{V})$. Note that both $0 \leq \delta_a, \delta_r \leq 1$, with greater value denoting more robustness.

## 4.1 Results and Analysis

We present the result of our benchmark on the baselines and analyse several interesting trends.

**Increasing occlusion severity over actor region reduces performance.** In Tabs2,3 , we occlude actor region with multiple occluders. It can be clearly observed that as the occlusion severity is increased, the performance of all the action-detectors is reduced. The notable thing is that VCAPS is most robust to severe occlusion (i.e. FG3) (in terms of absolute scores) on the larger O-UCF dataset (i.e. 36.9%) , and YOWO is most robust to occlusions on smaller JHMDB dataset (i.e. 46.2%). The larger performance difference in MOC and YOWO could be attributed to the treatment of action-detection as a box-regression problem in MOC[46]/YOWO[39] vs dense-semantic mask prediction in the VideoCapsuleNet[13]. Regressing the 4 coordinates of a box has a minimum probability of error as $\frac{1}{4} = 0.25$, whereas dense-prediction reduces this error to 1/n, where n is the no of pixels being predicted and $n >> 4$. As long as at least four pixels are predicted along the boundaries of the box, the method performs relatively well[3]. Also capsules have been shown to be extremely robust to occlusions on Multi MNIST[60], and we believe similar inductive bias extends to videos.

**Are backbones with more parameters *necessarily* more robust?** To answer this question, we tried CNN/Transformer-based backbones on two of our best performing methods, i.e. YOWO & VCAPS. In Table 5, it is evident that Mvitv2-S with 50% less parameters (35M) can outperform other 3D backbones like ResNext using large as 89M parameters (i.e. 67.3% on O-UCF & 86.8% on O-JHMDB). This shows that the internal attention mechanism in transformers *doesn't necessarily need more parameters* to improve robustness. Note that VCAPS with Mvitv2-S backbone (67.3, O-UCF) largely outperforms YOWO which also uses Mvitv2-S backbone (31.2, O-UCF). A crucial architectural difference is that the VCAPS-Mvitv2 uses 2 additional layers of capsules, whose qualitative effects shall be revealed in a later section.

---

[3]That is why predicting a dense mask and then fitting a rectangle around it during evaluation is better than directly regressing a box. (see supplementary.)

Table 4: **Effect of realistic occluder-motion on OVIS-UCF/OVIS-JHMDB datasets.** Models in general perform better on static occlusions than dynamic occlusions.

| | Backbone | OVIS-UCF | | OVIS-JHMDB | |
|---|---|---|---|---|---|
| | | Static | Dynamic | Static | Dynamic |
| MOC[46] | DLA34[79] | 14.8 | 12.4 | 26.8 | 31.9 |
| YOWO[39] | ResNext[74] | 16.5 | 10.1 | **37.6** | **35.3** |
| VCAPS[13] | i3D[7] | **32.3** | **21.7** | 20.9 | 17.3 |
| YOWO[39] | Resnet18[18] | 17.3 | 11.8 | **28.0** | **24.2** |
| VCAPS[13] | Resnet18[18] | **33.2** | **21.9** | 7.7 | 7.3 |

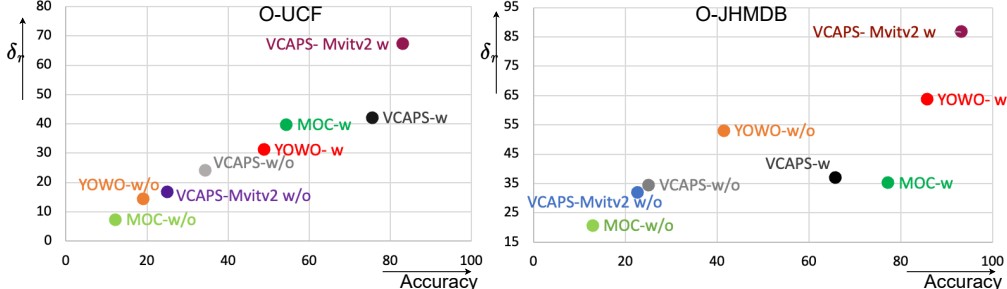

Figure 3: **Effect of pretrained weights:** VCAPS-Mvitv2 outperforms all models on both O-UCF and O-JHMDB. (X Axis:) Accuracy without occlusions. (Y Axis): Relative Robustness of Model. Top-Right corner of each plot corresponds to most robust model.

**Pre-training improves transformer's robustness to occlusions.** To investigate why transformers perform so well out of the box, we train them from scratch *without* pre-trained weights and test them on O-UCF/O-JHMDB datasets. From Fig 3 (i), it can be observed that transformers (i.e. VCAPS-Mvitv2) perform poorly *without* weights compared to other models. However, on training with Kinetics 400 weights, the transformers achieve state-of-the-art results. Therefore transformers are *not naturally more robust*, but pre-training helps to improves their intrinsic-robustness. This also reveals the glaring dependency of transformers on large amounts of data and compute for pre-training.

**Increasing parameter size in same model family improves occlusion robustness** By universal-approximation theorem [31], a larger neural net can learn better representations. We investigate whether similarly increasing the model parameter size (and hence internal-representational capacity) could lead to better occlusion robustness. In Fig 4(right), it can be seen that changing the backbones from Resnet 18→50→101 follows this trend. This shows that increasing parameter size in the *same* model family generally leads to improved occlusion robustness.

**Effect of occluder motion on backbones.** We experiment with realistic occluder motions by super-imposing occluders from OVIS datasets upon UCF/JHMDB videos, resulting in benchmark datasets called OVIS-UCF and OVIS-JHMDB respectively. In Tab14, we observe that in general models are performing better during static occlusions than dynamic occlusions. The differences still hold when the backbones are made consistent[19]. This shows that resolving actor location becomes difficult when both occluders and actors move together. Therefore, reasoning about temporal motion properly remains a worthwhile architectural pursuit. We had also experimented with *controlled* occluder motions like circle, sinusoids. For those results, we refer the reader to the supplementary.

From this section, we conclude that using transformers as a backbone and training *using* pre-trained weights can greatly improve robustness.

## 5 Improving Robustness under Occlusion

Next, we investigate if we can make these models robust against occlusion. Augmentation is a well studied technique to induce robustness in models against distribution shifts. We experimented with synthetic occlusions as augmentations to make the existing models robust. We generate augmented samples by superimposing occluders at random locations, various severity levels and motion. Training existing baselines on these augmented samples allows to improve occlusion robustness (Table 6).

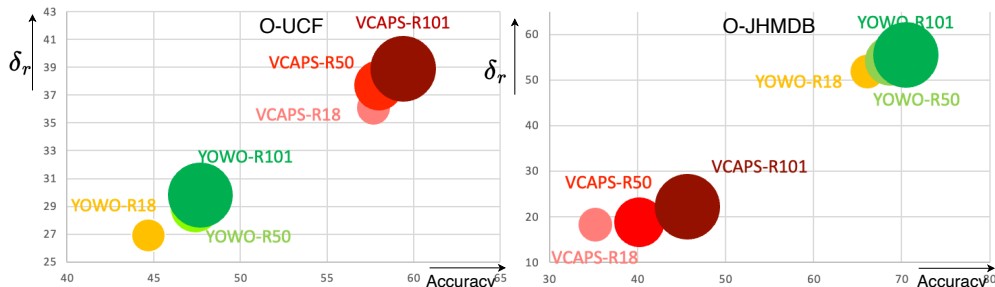

Figure 4: **Effect of number of model parameters:** (i) Increasing parameter size within the same model family yields more robustness. (X Axis:) Accuracy without occlusions. (Y Axis): Relative Robustness of Model. Top-Right corner of each plot corresponds to most robust model.

Table 5: **Effect of Backbones:** Transformers outperform other backbones in less than half the parameters. Clean evaluation is done on UCF-24 and JHMDB-21 test sets.

| Methods | Backbone | #Params (M) | O-UCF Clean | Occ | Clean | Real-OUCF Occ | O-JHMDB Occ |
|---------|----------|-------------|-------|-----|-------|------|------|
| YOWO[39] | Resnet18[18] | 11.7 | 44.7 | 26.9 | 4.4 | 66.1 | 51.8 |
| | YOLO + ResNext[74] | 89 | 48.8 | 31.2 | 7.6 | 85.7 | 63.7 |
| | MVITv2-S[45] | 35 | 69.4 | 48.2 | 9.1 | 81.3 | 68.2 |
| VCAPS[13] | Resnet18[18] | 11.7 | 57.7 | 36.1 | 6.4 | 35.2 | 18.3 |
| | ResNext[74] | 89 | 61.2 | 34.7 | 10.8 | 49.3 | 27.7 |
| | MVITv2-S[45] | 35 | **83.1** | **67.3** | **13.1** | **93.1** | **86.8** |

Next, we make some very surprising observations. In Table6, Swapping the backbone of one of the methods (VCAPS) with a transformer (i.e. Mvitv2) yields the highest performance across both UCF/JHMDB. A non-trivial observation is that a transformer- based network trained *without* augmentation outperforms all the other methods which are trained using *occlusion as an augmentation* (67.3 vs 49.9 on UCF). Training the transformers further with occlusion-based augmentation improves the performance from 67.3% to 81.6%. This is notable because this performance *on a occluded test set* (i.e. 81.6%) comes close to how the detector functions in the *absence* of occlusion,i.e. (83.1% on UCF). The remaining gap of 1.5% can be bridged by explicit occlusion modelling. Therefore, transformers trained with occlusions as a augmentation strategy can significantly improve robustness.

### 5.1 Importance of Capsules

Here, we explore the importance of capsules towards occlusion modelling. In Tab9a, we remove all the capsule layers from VCAPS-Mvitv2. Note that the architecture then reduces to a standard 3D Unet[10], and the performance drops from 67.3% to 64.1%. Therefore, capsules are a *crucial component* for the performance of VCAPS-Mvitv2. [4]

**Emergent Object/Occluder separation in capsules:** Next, we qualitatively explore the behaviour of capsule-based models towards occlusions. We choose our best-performing VCAPS-MvitV2 model which has never seen occlusions during training and feed-forward an occluded sample during inference in Fig5. The activation maps of some of the capsules in the primary layers have been visualized. Each capsule has an activation map of size H×W, where each value denotes the confidence about the location of a particular entity (object). Applying a threshold of 0.7 results in a binary mask. It can be seen that some of the capsules selectively look at the objects (i.e. actors), and other capsules focus on occluders (eg, chair, sofa). Note that the capsules have never seen occluders during training, but are still able to focus on them. Therefore, architectural constraints like binding[17] object-specific pixels to particular slots (eg, capsules)[36] allows behaviour like unsupervised object discovery to emerge[49].

---

[4]Quantitative improvements are not observed if the number of capsule layers are increased further than two. One reason is that capsules were originally invented for rigid bodies [23, 30] and video-capsules would need to enforce dynamic deformable reference frames.[63, 53]/ learn separate-canonical frames in case of multiple instances.[78]

Table 6: **Effect of Augmentation:** Transformer based backbones outperform other CNN based models which have even used occlusion as an augmentation. Clean evaluation is done on UCF-24 and JHMDB-21 test sets. w/o: without augmentation, w: with augmentation.

| Methods | Backbone | Type | O-UCF | | | Real-OUCF | | O-JHMDB | | |
|---|---|---|---|---|---|---|---|---|---|---|
| | | | clean | w/o | w | w/o | w | clean | w/o | w |
| MOC[46] | DLA34[79] | CNN | 54.4 | 39.8 | 44.1 | 3.1 | 5.8 | 77.2 | 35.3 | 55.9 |
| YOWO[39] | YOLO[56]+ ResNext[74] | CNN | 48.8 | 31.2 | 44.9 | 7.6 | 9.0 | 85.7 | 47.8 | 68.0 |
| VCAPS[13] | i3D[7] | CNN | 75.5 | 42.2 | 49.9 | 9.9 | 10.8 | 65.7 | 35.6 | 59.7 |
| VCAPS[13] | Mvitv2[45] | Transformer | **83.1** | **67.3** | **81.6** | **10.5** | **12.1** | **93.1** | **86.8** | **90.5** |

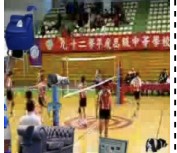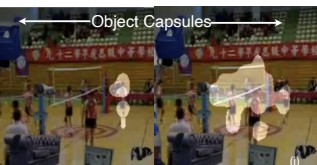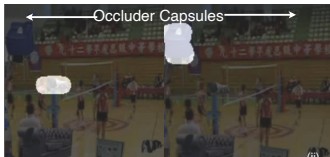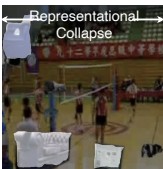

Figure 5: **Emergent object/occluder separation in capsules.** (top left) An occluded video is feed-forwarded through VCAPS-Mvitv2 and activations of primary layer capsules are visualized. Capsules can (i) parse an actor into constituent body parts without instance-level supervision[22, 28] (ii) segment multiple actors & objects of action (volleyball net). (iii) show evidence of focusing on occluders never seen during training (iv) **undergo representational collapse**, where a single capsule starts representing multiple objects (wholes) when number of objects in the scene are greater than number of capsules in the network. Best viewed in color.

**Representational collapse:** Each capsule is meant to represent one entity (object) only. If there are more objects (i.e. occluders) in a video than the capsules in a network, a single capsule gets forced to encode multiple disjoint objects together. This problem can't be solved without increasing the number of capsules further and *retraining* the network again[5] This similar issue arises in the standard object detection literature where the maximum number of objects that a network can detect is equal to the number of object proposals it was originally trained with [5, 9, 57].

### 5.2 GLOM: Islands of agreement *can emerge* from token representations.[26]

We qualitatively explore the effectiveness of our VCAPS-Mvitv2 model presented in Tables7,9b. Specifically, we assume that each of the output token representation from Mvitv2 is equivalent to a column-based activity vector [26]. We perform t-sne reduction [64] of each vector to three dimensions, and linearly project obtained components to the RGB range of $[0, 256)$ in Fig6. Note that obtained maps are not simple patch-based correlations among token-representations[52], but rather lower dimensional clustering of vectors themselves plotted as identical colors[81]. Furthermore, our VCAPS-Mvitv2 has *only* received actor-level annotations/ action-label supervision and *not* any form of occluder masks [83], contrastive-textual supervision[68] or distilled knowledge[6] during training. The only used prior is Kinetics pretrained weights learnt from action-recognition.

This suggests that the lower layers in our VCAPS-Mvitv2 (i.e. transformer layers) perform semantic grouping of pixels into objects[75] (object discovery). Next, higher layers (capsule layers) selectively segregate objects/occluders into instance-specific slots 5. Thus transformers and capsules *complement* each other to achieve first two stages of the binding process[17]. Finally, our decoder reasons to suppress occluder-specific features and focus/infill on actor-centric regions during localization, i.e. Fig 7. From this section, we conclude that two capsule layers stacked on top of transformer-based backbones can help improve occlusion-robustness. The neuro-symbolic advantage of capsules

---

[5]This differs from how the brain iteratively parses variable number of objects in a *constant* number of neurons, although it may take more time[50]. In contrast, parallel models of attention[65] and capsules[60] have assumed singular fixation. This requirement *forces* object queries to compete through self attention[5] for explaining perceptual parts of an input[49]. However, competing queries raise memory requirement linearly for each object that needs to be decoded[5]. Therefore, while neural nets can *encode* in a *finite* number of learnable neurons, parallel decoding *suffers* a fundamental memory bottleneck[5, 8, 67] (that can't be solved by retraining at scale). Surprisingly, neural fields *aren't constrained by this limitation* because they enforce a *single* world-specific coordinate frame instead of separate object-specific canonical frames.[80, 78]

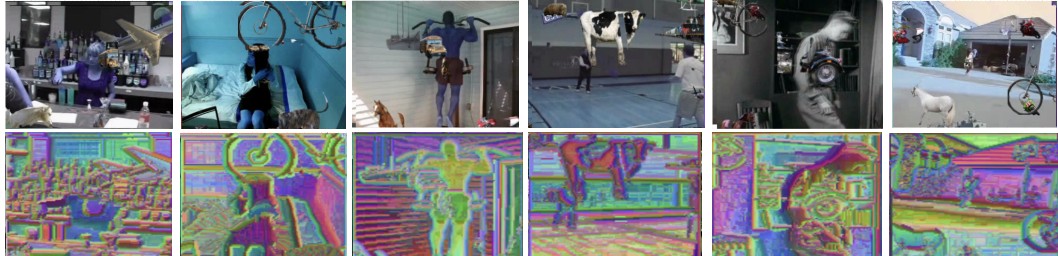

Figure 6: **Emergent islands of agreement on O-JHMDB dataset[15]**: Output token representations of our VCAPS-Mvitv27 can be thought of as column-based vectors[26, 4]. All the vectors belonging to the same object should agree among themselves. Reducing dimensionality of these vectors to three using t-sne shows evidence of such islands of agreement as belonging to identically-colored regions. Note that obtained islands show both occluders as well as actors. Our video-based model is only trained with actor-level annotation/action class labels and no occluder-specific masks.

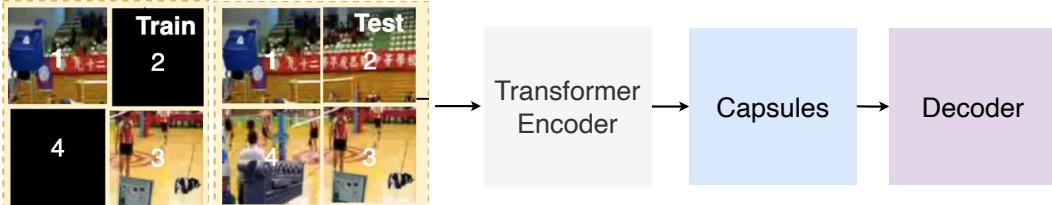

Figure 7: **Proposed Token Masking** (from left to right) An input video is partitioned into tokens and fed to a transformer. (ii) During training, a random number of tokens are blacked out. (iii) During inference, all the tokens are switched on and the actor is localized under occlusions.

Table 7: **Token Masking**: Improves performance of models trained even using occlusion as augmentation.(D)- dropout on intermediate network layers. (T)- proposed token masking

| | Backbone | Mode | O-UCF | | | O-JHMDB | | |
|---|---|---|---|---|---|---|---|---|
| | | | clean | aug | aug+mask | clean | aug | aug+mask |
| MOC[46] | DLA34 [79] | D | 54.4 | 44.1 | 47.4 | 77.2 | 55.9 | 61.3 |
| YOWO [39] | ResNext | D | 48.8 | 44.7 | 46.8 | 85.7 | 68.0 | 72.8 |
| VCAPS[13] | i3D [7] | D | 75.5 | 49.9 | 51.5 | 65.7 | 59.7 | 62.5 |
| VCAPS | Mvitv2 [45] | D | **83.1** | **81.6** | 81.9 | **93.1** | **90.5** | 91.7 |
| VCAPS | Mvitv2 [45] | T | - | - | **82.2** | - | - | **92.4** |

combined with innate natural robustness of transformers forms the basis of our best model, i.e. VCAPS-Mvitv2.

### 5.3 Token masking

Till now, we have seen that architectures possessing a transformer based backbone with a few capsule layers on top are highly robust to occlusions. Now, we propose token-masking to improve their innate robustness. Given a video of dimensions $T \times H \times W$, a transformer creates $L$ spatio-temporal patches each of which is projected to a common dimensionality $D$ during input. Our idea is that randomly masking (blacking out) some of the tokens during training *introduces* additional occlusion robustness in the model. Mathematically, we generate L iid bernoulli random variables

$$M_i \sim Bernoulli(p) \ \forall i \in [1, L] \tag{1}$$

where $p$ is the masking probability. Each $M_i$ is repeated $D$ times, so that a particular token $P$ is fully masked before input. Next, $M_L^D$ is multiplied element wise with the input sequence to give final masked input $I' = I \odot M$ where $I,M$ are input sequence vector and masking vector of dimensions $\mathbb{R}^{L \times D}$ respectively, and $\odot$ is element-wise hadamard operation. During inference, none of the tokens are masked, and the input occluded video sequence is directly feed-forwarded through the network.

**Quantitative Results** In Table 7 it can be seen that such token-masking improves *even* the performance of models which have used occlusion as a data augmentation from 81.9% to 82.2% on

Table 8: **Comparison with existing methods:** Comparison of our method across existing supervised approaches, *: denotes results using a CSN152 backbone.

| | Backbone | | UCF-24 | | | JHMDB-21 | | |
| | 2D | 3D | f-mAP | v-mAP | | f-mAP | v-mAP | |
| Methods | | | 0.5 | 0.2 | 0.5 | 0.5 | 0.2 | 0.5 |
|---|---|---|---|---|---|---|---|---|
| *Yang et al.*[37] | ✓ | | 75.0 | 76.6 | - | - | - | - |
| *Li et al.* [46] | ✓ | | 78.0 | 82.8 | 53.8 | 70.8 | 77.3 | 70.2 |
| *Kopuklu et al.*[39] | ✓ | | 80.4 | 75.8 | 48.8 | 75.7 | 88.3 | 85.9 |
| *Zhao et al.*\*[86] | | ✓ | 81.3 | 85.3 | 60.2 | 82.3* | 81.8 | 80.7 |
| *Duarte et al.*[13] | | ✓ | 78.6 | 97.1 | 80.3 | 64.6 | 95.1 | - |
| *Kumar et al.*[42] | | ✓ | 69.2 | 95.3 | 71.9 | 68.1 | 96.8 | 68.4 |
| *Tao et al.*[73] | | ✓ | **83.7** | - | - | 86.7 | - | - |
| Ours | | ✓ | 81.2 | **98.6** | **83.1** | **93.0** | **98.1** | **92.8** |

Table 9: **Ablations and evaluation** of proposed token-masking on Realistic Occlusions.

(a) Capsules stacked on transformer backbone improve occlusion robustness. Testing is done on O-UCF.

| | w/o caps | w/ caps |
|---|---|---|
| VCAPS (MvitV2) | 64.1 | **67.3** |

(b) **Realistic Occlusions:** Models trained with token-masking outperform other models on Real-OUCF. †-Training with occlusion augmentation & dropout. ‡-Training with occlusion augmentation & token-masking.

| | MOC[†] | YOWO[†] | VCAPS[†] (i3D) | VCAPS[‡] (Mvitv2) |
|---|---|---|---|---|
| Occ | 7.2 | 9.6 | 11.7 | **14.3** |

O-UCF and 91.7% to 92.4% on O-JHMDB. Note that for VCAPS(Mvitv2), masking some of the tokens during training performs better than dropping some of the intermediate network neurons. This observation is in line with how learning objectives similar to pixel-level reconstruction in masked image modelling (MIM)[20] yields better self-supervised representations.

**Comparison With existing Methods:** We compare with existing methods on f-mAP and v-mAP. In Tab8, our method obtains 83.1 vmAP on UCF-24 thereby indicating most localization robustness. On JHMDB-21, we obtain 92.8% in terms of the absolute v-mAP score, which is significantly better than other existing methods. We acknowledge that TubeR[86] and ST-Mixer [73] are slightly better than our method in terms of f-mAP scores on UCF-24 dataset.

**Realistic Occlusions:** Furthermore, we evaluate our *baseline* models and the token-masked models on the Real-OUCF dataset. In table 6, we observe that the models trained with occlusion as augmentation perform *better* than models without augmentation. As evident in Table 9b, models trained with our token-masking approach also outperform other models in *realistic* scenarios.

## 6   Conclusion

We have conducted the *first-ever* benchmark study to evaluate the impact of occlusions in spatio-temporal action detection. This study provides several interesting key insights including, i) Models perform better on static occlusions vs dynamic occlusions in *realistic scenarios.* ii) Transformer-based models possess greater natural robustness compared to other models using occlusion as data augmentation. iii) Pre-training improves robustness of transformers more than CNN models. (iv) Robustness can be improved further by leveraging components like capsules and training jointly with occlusions as data augmentation. (v) Recipes like simply masking some input tokens of transformers during training can introduce additional robustness and obtain state-of-the-art performance under realistic occlusions. Our benchmark, datasets and code have been released at this link.

**Limitations:** We have only focused on annotating visible regions of the occluded-objects in line with official COCO protocol[47]. Another direction could also focus on predicting *missing* regions, i.e. amodal-segmentation. Classically, semantic-segmentation has assumed that one pixel can belong to only one object[19]. However, this is not true for occlusions. Inductively, this symmetry issue has been resolved in works like Maskformer2[9] where multiple objects per pixel can be predicted by removing the softmax assumption. We note that there is still a significant gap left to bridge in existing models for improving robustness to *realistic* occlusions9b.

# 7  Acknowledgement

This research is based upon work supported in part by the Office of the Director of National Intelligence (Intelligence Advanced Research Projects Activity) via 2022-21102100001 and in part by University of Central Florida seed funding. The views and conclusions contained herein are those of the authors and should not be interpreted as necessarily representing the official policies, either expressed or implied, of ODNI, IARPA, or the US Government. The US Government is authorized to reproduce and distribute reprints for governmental purposes notwithstanding any copyright annotation therein.

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

# 8 Appendix

## A1. Broader Impact Statement

A decade ago[6], [82] showed that successive filters of a convnet could act as general forms of edge, shape and texture detectors. In Fig 6, we now illustrate that higher layers of a neural net can learn to group pixels into objects at a semantic level without any explicit localization supervision/ multi-modal alignment . This shows that we could learn non-parametric object-queries at the highest levels in the architecture (via unsupervised-clustering), instead of directly injecting learnable parameters/proposals into lower decoder-layers[5]. In nature, there can be a large number of objects present in the retinal frame (eg, leaves of a tree). Depending on the granularity of fixation (eg, separation between multiple trees), we might care only about a subset of those classes[47]. However, the objects we don't care about still exist, even though an object query might not bind to them. Therefore, the encoding process in a neural net should *not* prevent separate islands of finer objects (eg, leaves) from getting formed [7].

The representational collapse issue observed in Sec5.1 presents a memory-scaling bottleneck. One way to get around it is to consider the behaviour of a self-replicating cellular-automaton [71] like GLOM[24]: unfolding a singular embedding creates dynamic connectionist hardware on-the-fly[29]. This would also allow us to spiritually succeed the computationally-expensive distillation setup, i.e. simply copying a singular embedding lesser number of times on a weaker hardware would allow the student model to exist. A key question that thus remains unanswered for our community is how to compress the entire knowledge of a neural-net into this singular entity (eg, biological seed) and the mechanism behind its unfolding[35](which is an inverse of the protein-folding problem).

## A2. Mortal computation requires self-replicable embeddings

On the other hand if we want to pack similar levels of intelligence in a brain-like interface which consumes less than 25 watts daily, we have to take an alternate biologically-plausible route[26]: intelligence can be encoded in a single cell (embedding) much like how the genetic code of a human gets encoded in the DNA. In nature, multiple copies of a zygote (cell) lead to emergence of an animal's internal organs. Similarly, in our computers multiple copies of this single replicable embedding shall lead to networks whose structure gets discovered 'on-the-fly' according to local hardware constraints.

It has been well established in distillation literature that [27] a higher-parameterized teacher network can teach a lower-parameter student network to obtain similar performance. If all the knowledge of the teacher could be compressed into a singular embedding, then we won't need distillation at all. Simply copying the single embedding many times on a weaker student hardware would be enough to allow the student model to exist. Although the student model would be weak due to lesser copying, the replication step for both student/teacher stems from a common learnt singular representation[8]. These 'connectionist-networks'[29] which start their existence from a single embedding and only remain in the memory as long as the hardware is powered on (hence the name mortal) would require learning algorithms other than backpropogation: where the parameters of each layer could be updated without knowing the precise feed-forward mathematical-functions[25] and allow dynamic growing of the network on a smaller hardware[34]. We remain optimistic for the future that such mortal computation offers to humanity[25].

## A3. Additional Supplementary Material

This section discusses the supplementary materials in addition to the main manuscript. The supplementary material contains seven sections:

- §A4. proposes a *new task* called action-segmentation involving *instance-level localization* of actor in a video and presents a benchmark to streamline further research in the field.

- §A5. explores the *importance of background context* in action detection.

- §A6. presents the full benchmark and analyzes the background bias property in existing detectors.

---

[6]This section is meant to be philosophical and optimistic in nature.

[7]And instead encode the complete part-whole hierarchy of a scene[12]. This shall resolve the issue of oversegmentation/ inability to distinguish between part-wholes that SAM[38] still faces when given a grid of input point prompts at a very-fine granularity.[38]

[8]An undeniable fact of nature is that intelligence/consciousness in humans emerges from self-replication of a singular cell (zygote). It still remains to be seen whether singular prokaryotic organisms like amoeba themselves are conscious [55, 43]

- §A7. analyzes our results for *synthetic* occluder motions on O-UCF & O-JHMDB datasets.

- §A8. discusses more detail about the video-collection and annotation process of our curated Real-OUCF dataset.

- §A9. presents some qualitative samples from the proposed three Benchmark datasets, along with UCF-101 instance-level annotations. All the datasets, benchmarks and codes for this work will be released for free public usage at `https://anonymous.4open.science/r/OccludedActionBenchmark-B9E2`.

- §A10. provides the NeurIPS recommended datasheet explaining the dataset collection mechanism and other important details.

## A4. A New Instance Level Benchmark

Traditionally[46, 86], spatio-temporal action-detection has relied on predicting *bounding boxes* across an actor for *each* frame. A much harder task instead would be a *finer-grained* localization, i.e. predicting instance-level mask for an actor instead of only bounding boxes. Surprisingly, to the best of our knowledge, only one approach i.e. VideoCapsuleNet[13] is able to solve the much harder task of instance-level action segmentation by adapting the network trivially out-of-the-box.

We argue that research in the field of instance-level action-segmentation has largely been inhibited due to the lack of proper instance-level actor annotations for standard action-detection datasets[61]. While the popular JHMDB[33] dataset consists of instance puppet masks, the larger UCF-24 dataset only has bounding box annotations. To rectify this, we release the instance-level annotations for UCF-24 some of whose samples have been illustrated in Fig1213. Our insight is that these annotations will now allow the existing research in action-detection and Video Instance Segmentation [72] to progress concurrently due to inherently similar problem formulations [9]. Finally, we note that the results of VCAPS on instance-level benchmark in Tab10 are significantly lower than on the bounding-box level benchmark in Tab12. This indicates that instance-level localization task is significantly harder task than isolating 'broader level' bounding boxes.

**Annotation Procedure for UCF-24:** We generate instance-level segmentation masks on UCF101-24 videos leveraging the recent SOTA in Video Instance segmentation [72]. To make sure that our instance-level action tubes are temporally coherent, we perform inference over successive chunk sizes of 100 frames, with a temporal overlap of t = 30 frames. Next, we run a sliding window for smoothening the predicted tube to remove any stray pixel-level artifacts. Finally, we manually refine the obtained segmentation masks using the CVAT tool.[1]

Table 10: **Instance Level Benchmark on O-UCF**: A benchmark showing a VCAPS model trained using our instance-level annotations on UCF-24 datasets and evaluated on clean/occluded test-set.

| | Occ As Aug | Clean | BG1 | | | BG2 | | | BG3 | | | Circle | Sin | Avg |
|---|---|---|---|---|---|---|---|---|---|---|---|---|---|---|
| | | | FG1 | FG2 | FG3 | FG1 | FG2 | FG3 | FG1 | FG2 | FG3 | | | |
| VCAPS[13] | × | 34.7 | 29.3 | 21.0 | 13.7 | 28.3 | 19.1 | 13.5 | 29.2 | 20.0 | 13.1 | 13.3 | 18.1 | 19.9 |
| VCAPS[13] | ✓ | 41.1 | 36.4 | 29.5 | 23.5 | 36.3 | 28.1 | 24.4 | 34.9 | 28.2 | 23.8 | 22.6 | 25.8 | 28.5 |

## A5. Preliminary Experiments

In Fig1 of the main manuscript, we had run a set of preliminary experiments which served as a motivation for this benchmark study. One notable result has been illustrated in Table 11. *Distractor* refers to the setting when there is just one single occluder in the background of a video. *No Context* refers to the setting when all the background pixels of the video have been blackened, thereby making it easier for the network[46, 13] to classify the remaining pixels as an actor. It can be clearly observed that the highest drops (i.e. $\delta_r$ for No Context case) are observed when the background is entirely masked. This shows that existing networks are highly *biased* to the background information while trying to reason about an actor location. We note that this dependency on background is *counter-intuitive* to the desirable behaviours of learning object(actor)-centric representations[5, 86].

Table 11: **Distractor vs Background-Context Sensitivity**: Highest performance drops are observed when background is masked (no context), whereas presence of distractor objects in background has slightly less effect.

| Methods | UCF-24 | | | | | JHMDB-21 | | | | |
| | Clean | Distractor | | No Context | | Clean | Distractor | | No Context | |
| | | Abs | $\delta_r$ | Abs | $\delta_r$ | | Abs | $\delta_r$ | Abs | $\delta_r$ |
|---|---|---|---|---|---|---|---|---|---|---|
| MOC[46] | 54.4 | 38.6 | 0.71 | 26.6 | 0.49 | 77.2 | 53.7 | 0.70 | 49.5 | 0.64 |
| YOWO[39] | 48.8 | 43.8 | **0.90** | 32.2 | 0.66 | **85.7** | **73.5** | 0.86 | **66.5** | **0.78** |
| VCAPS[13] | **75.5** | **67.7** | **0.90** | **53.1** | **0.70** | 65.7 | 61.8 | **0.94** | 47.5 | 0.72 |

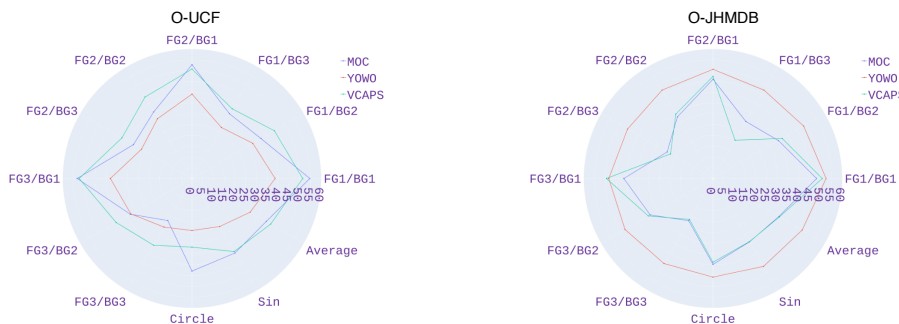

Figure 8: **Occlusion Robustness** across multiple severity levels. The outermost envelope represents most robust model. Radial Axis represents vmAP-0.5 score. For O-UCF, VCAPS is most robust, whereas for O-JHMDB, YOWO is most robust across several categories.

Table 12: **O-UCF Benchmark:** Illustrates the full benchmark across 9 severity levels of static occlusions and different trajectories in dynamic occlusions.

| Methods | Occ As Aug | BG1 | | | BG2 | | | BG3 | | | Circle | Sin | Avg |
| | | FG1 | FG2 | FG3 | FG1 | FG2 | FG3 | FG1 | FG2 | FG3 | | | |
|---|---|---|---|---|---|---|---|---|---|---|---|---|---|
| MOC[46] | ✗ | 54.7 | 37.0 | 34.7 | 52.7 | 35.6 | 31.4 | 53.3 | 33.1 | 22.5 | 42.9 | 39.9 | 39.8 |
| YOWO[39] | ✗ | 38.6 | 32.6 | 27.3 | 39.1 | 32 | 27.1 | 37.9 | 32.8 | 26 | 24.1 | 25.7 | 31.2 |
| VCAPS[13] | ✗ | 51.4 | 44.2 | 37.3 | 50.7 | 43.6 | 37.6 | 52.4 | 40.7 | 35.7 | 31.8 | 39.1 | 42.2 |
| MOC[46] | ✓ | 48.3 | 43.6 | 39.6 | 47.3 | 43.1 | 39.1 | 47.2 | 44.6 | 38.5 | 48.1 | 45.6 | 44.1 |
| YOWO[39] | ✓ | 46.5 | 45.3 | 43.4 | 46.1 | 45.3 | 43.4 | 46.5 | 45.7 | 44.3 | 43.8 | 43.6 | 44.9 |
| VCAPS[13] | ✓ | 54.7 | 51.1 | 48.7 | 54.8 | 50.4 | 47.6 | 54.2 | 50.7 | 47.8 | 43.4 | 45.3 | 49.9 |

Table 13: **O-JHMDB Benchmark:** Illustrates the full benchmark across 9 severity levels of static occlusions and different trajectories in dynamic occlusions.

| Methods | Occ As Aug | BG1 | | | BG2 | | | BG3 | | | Circle | Sin | Avg |
| | | FG1 | FG2 | FG3 | FG1 | FG2 | FG3 | FG1 | FG2 | FG3 | | | |
|---|---|---|---|---|---|---|---|---|---|---|---|---|---|
| MOC[46] | ✗ | 48.1 | 35.1 | 30.5 | 46 | 33 | 24.6 | 41.4 | 33.7 | 22.6 | 39.8 | 33.9 | 35.3 |
| YOWO[39] | ✗ | 52.5 | 48.6 | 47.3 | 50.6 | 47.3 | 45.8 | 48.4 | 47.2 | 45.5 | 45.7 | 47 | 47.8 |
| VCAPS[13] | ✗ | 50.7 | 37.1 | 20.5 | 47.3 | 34.5 | 22.8 | 49.5 | 34.7 | 21.9 | 38.8 | 33.8 | 35.6 |
| MOC[46] | ✓ | 59.6 | 55.9 | 53.3 | 58.9 | 54.3 | 52.9 | 58.5 | 54.8 | 52.2 | 57.2 | 57.2 | 55.9 |
| YOWO[39] | ✓ | 71.1 | 70 | 69.3 | 69.2 | 70.3 | 66.9 | 68.4 | 67 | 65 | 65.7 | 65.1 | 68 |
| VCAPS[13] | ✓ | 60.8 | 59 | 54 | 61.2 | 56.9 | 53.9 | 61.5 | 57.8 | 54.4 | 55.3 | 58.2 | 59.7 |

## A6. Full Benchmark Analysis

In Tables 12, 13, we present the full benchmark of the proposed O-UCF and O-JHMDB datasets across the 9 severity levels of static occlusions and circular/sinusoidal dynamic motions.

---

[9]The only difference is VIS involves instance-classification which could be solved by just a single-frame object detection. However, action detection requires higher-level action-classification which requires temporal reasoning. The per-frame localization task of both the problems is fundamentally identical.

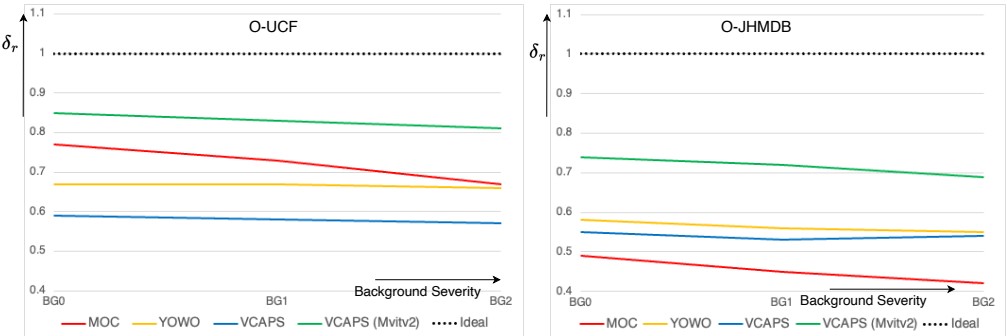

Figure 9: **Background Bias:** As the occlusion severity over background is increased, the performance of an ideal detector should not drop (dashed line). Our VCAPS-Mvitv2 shows highest robustness of all studied models. (X axis): Different severity levels of background occlusions. (Y Axis): Relative Robustness ($\delta_r$)

Table 14: **Static vs Motion Trends**: 2D backbones work well on Static Occlusions whereas 3D backbones work best under Dynamic Occlusions. S- Static Occluders. D- Dynamic Occluders. For analysing clear effects, results are shown with only one occluder over an actor (no bg occlusion).

| Methods | Backbone | Type | UCF-24 | | | | JHMDB-21 | | | |
| --- | --- | --- | --- | --- | --- | --- | --- | --- | --- | --- |
| | | | Clean | S | D | D-S | Clean | S | D | D-S |
| MOC[46] | DLA34 [79] | 2D | 54.4 | 28.7 | 27.3 | -0.4 | 77.2 | 34.4 | 34.0 | -1.1 |
| YOWO[39] | YOLO +ResNext | 2D+3D | 48.8 | 24.0 | 25.2 | 1.2 | 85.7 | 54.6 | 55.2 | 0.6 |
| VCAPS[13] | i3D [7] | 3D | 75.5 | 42.5 | 46.6 | 4.1 | 65.7 | 23.0 | 23.9 | 0.9 |

**Background Bias:** For an ideal action-detector, it is expected that occlusions in the background wont impact the localization performance in the actor region. We observe this trend in Fig9, where the performance of an ideal detector would be a black line parallel to x-axis. In the graph, most of the methods suffer negligible drops on increasing the occlusion severity in the background. Note that MOC[46] suffers most drop across different severity levels. Our VCAPS-Mvitv2 obtains the highest robustness across all occlusion severity levels for both O-UCF & O-JHMDB datasets.

## A7. Motion Analysis

In the main manuscript, we had experimented with *realistic* motion on the OVIS-dataset. We had also experimented with *synthetic* motions on O-UCF and O-JHMDB datasets, and show those results in Table 14. We observe that in case of MOC using the 2D DLA34 backbone, the performance is better in case of static occlusions, (i.e. 28.7 vs 27.3 on O-UCF). This shows that 2D backbones work well if occluder is static. On the other hand, YOWO uses a combination of 2D backbone (YOLO) and a 3D backbone (ResNext). This shows a slight positive improvement of 1.2% in UCF and 0.6% in JHMDB, thereby indicating that 3D backbone seems to be helping. The best improvement is evident by purely switching to a 3D based backbone as in the case of VCAPS, where gains as much as 4.1% can be observed. From this, it can be seen that 3D backbones *might* help networks reason about actor locations even under the challenging scenarios when the occluders might be moving. However, we note that these differences are close within a small margin of error. Also, our experiments with different type of trajectories in Tab1213, i.e. circle and sinusoids show inconsistent- trends thereby indicating that there is no particular motion type to which networks are *more* sensitive.

## A8. Real-OUCF dataset

We have curated Real-OUCF dataset for realistic occlusion scenarios.

**Video Selection Process:** The original videos in UCF-24 dataset were mostly of sporting events, but were of significantly lower resolution, i.e. 240 by 320. Furthermore, they consisted of only a single actor. Now, we have significantly upgraded that test set to reflect multiple actors which mutually occlude each other with a high degree of overlap as high as 99%. First, we scraped the videos matching keywords like "riding bike" etc. One curious way we were able to get such overlap ratios

was by searching specifically for events like *tandem surfing, tandem diving etc.* In tandem-events, two actors try to synchronize with each other. A lateral captured viewpoint, which captures multiple actors (with one actor behind another), offers very challenging yet realistic conditions for occlusions. Finally, our curated videos are consisting of professional sporting events like Olympics, as well as casual settings, for eg, people just playing cricket in a park.

**Annotation Criteria:** In spatio-temporal action detection, there are two questions that the annotations try to answer 1) what is the time interval during which the action occurs 2) what is the spatial location of the actor in *each* frame of the action-interval. Therefore, we first temporally crop the scraped videos to obtain action start and end times. Finally, for all the frames in between this interval, we spatially localize the actors using the CVAT annotation tool. One subtle thing is that action-detection does not try to label *all* the actors in the video. For eg, if some people are standing , and some people are doing some useful activity like biking, we are only concerned with the biking. This is in line with the annotation procedure of official UCF24.

## A9. Dataset Samples

In this section, we show some qualitative samples from our proposed-datasets and benchmarks. Specifically, Fig1011 shows the realistic occlusion dataset which we have collected for evaluating robustness to real-world occlusions. Next, Fig1415contains the synthetic dataset samples from O-UCF and O-JHMDB consisting of controlled occlusions. Fig1213 contains the exhaustive instance-level annotations of UCF-24 videos we have released to facilitate our proposed benchmark of action-segmentation.

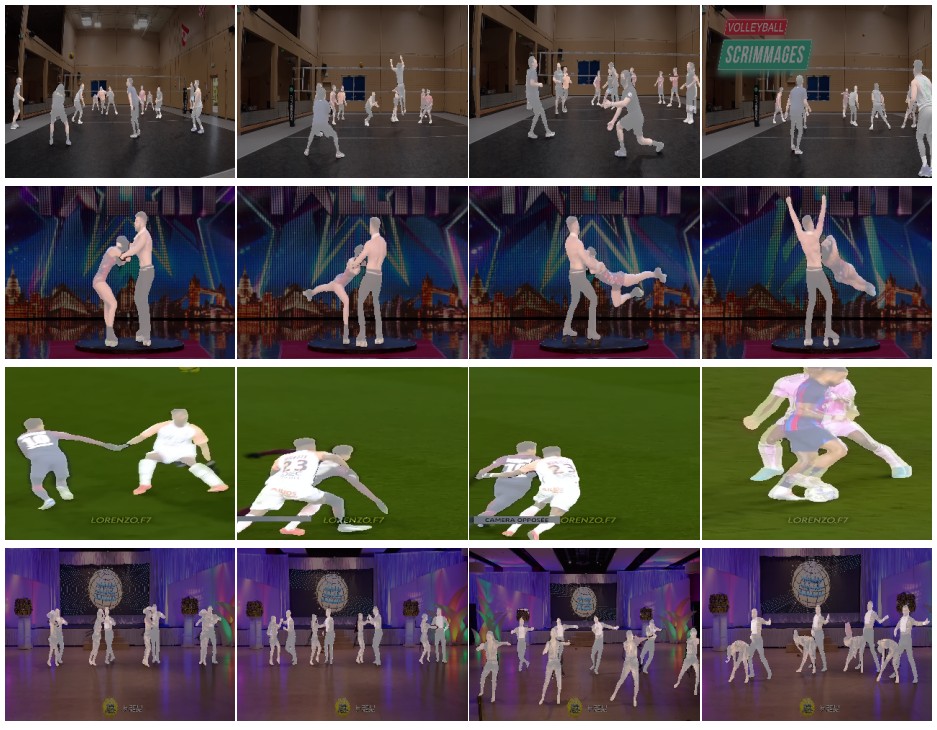

Figure 10: **Our Real-OUCF:** contains realistic occlusions with instance-level action annotations.

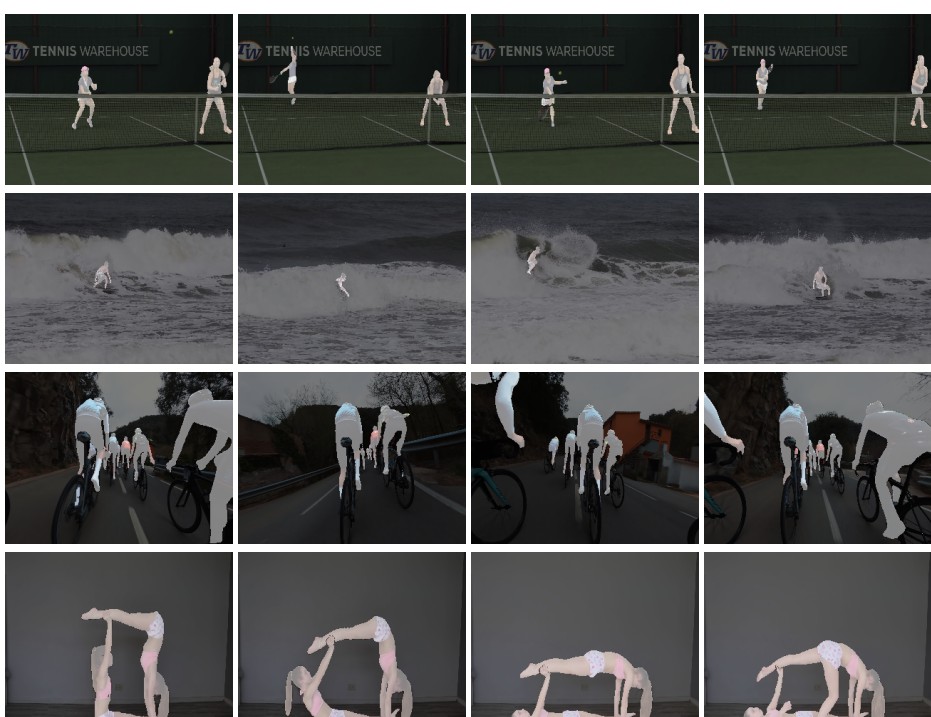

Figure 11: **Our Real-OUCF:** contains realistic occlusions with instance-level action annotations.

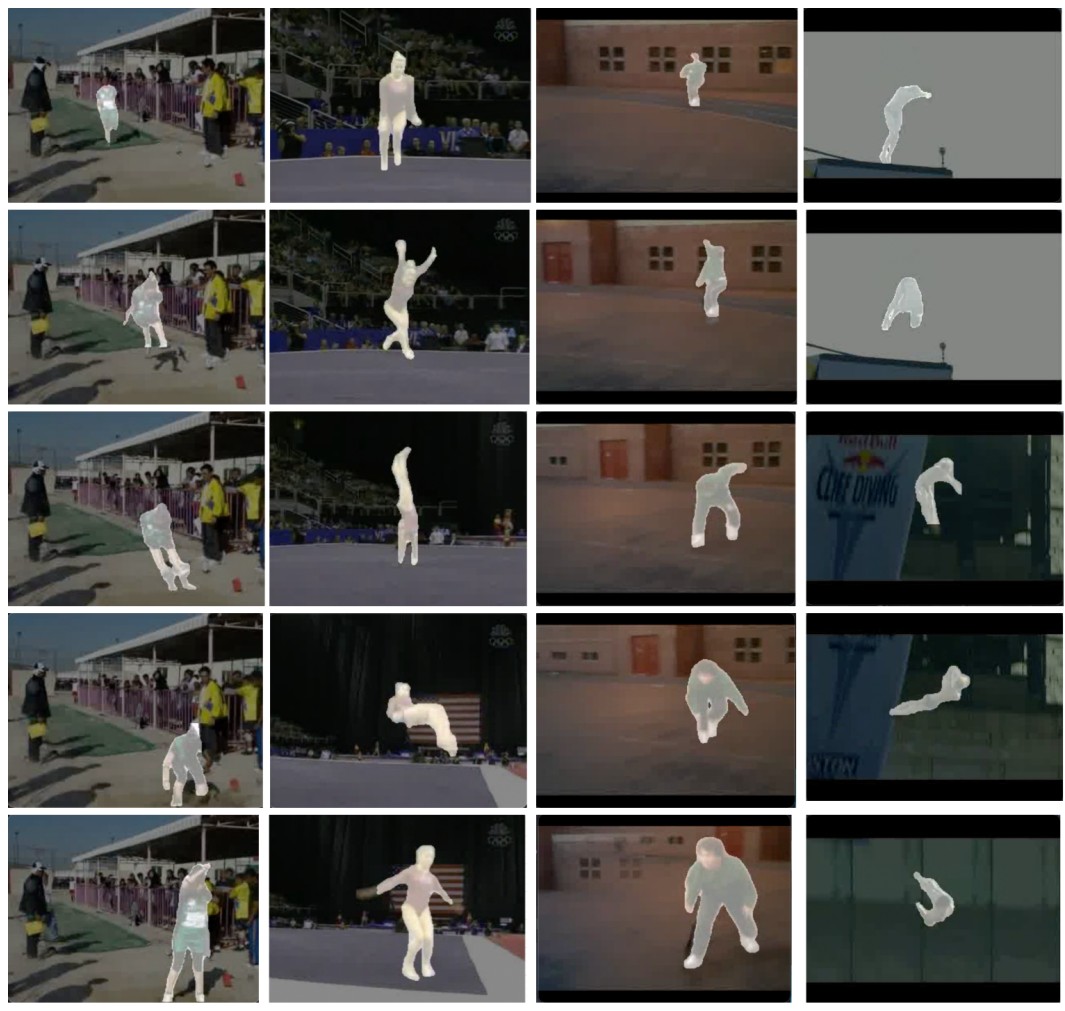

Figure 12: **Our instance-level annotations for UCF-24:** We propose a new benchmark of instance-level action segmentation and release official annotations.

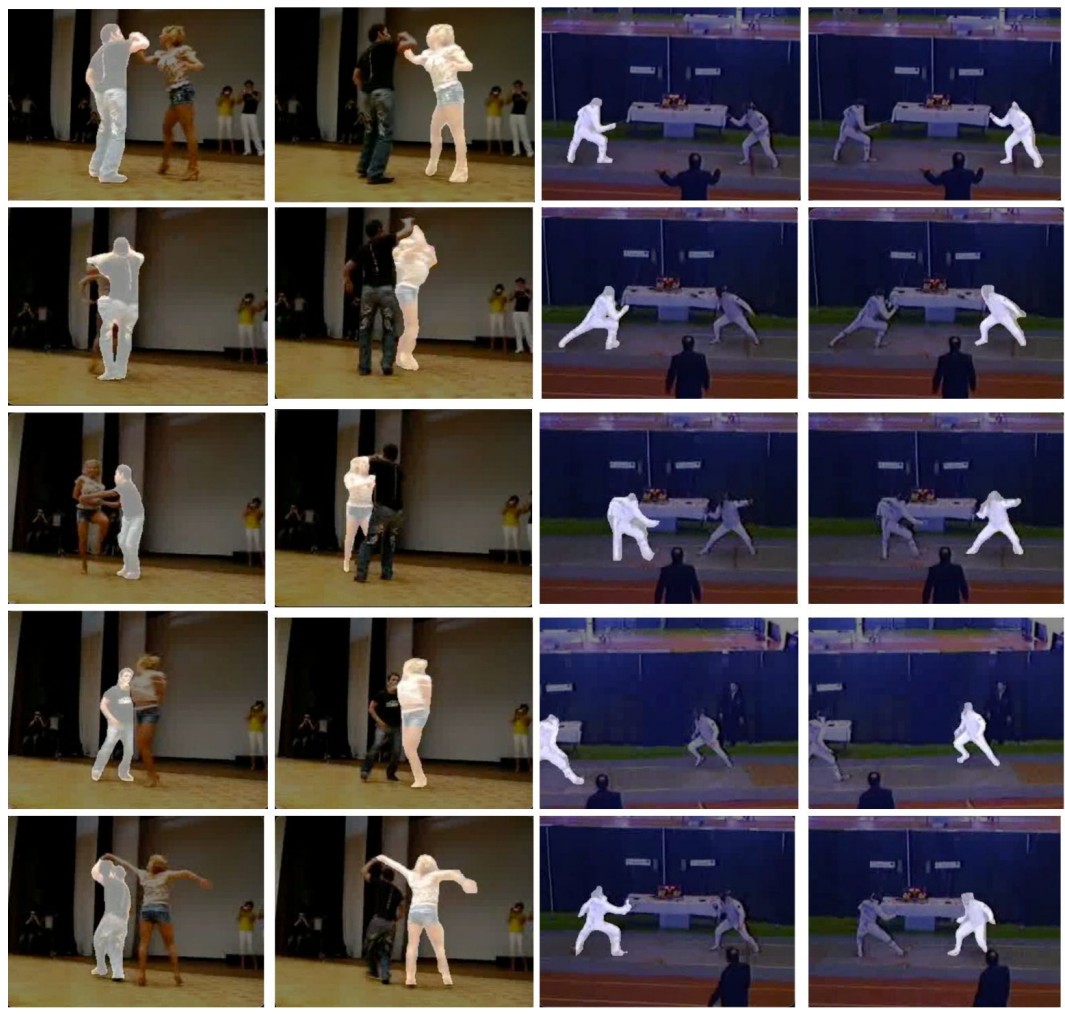

Figure 13: **Our instance-level annotations for UCF-24:** We propose a new benchmark of instance-level action segmentation and release official annotations.

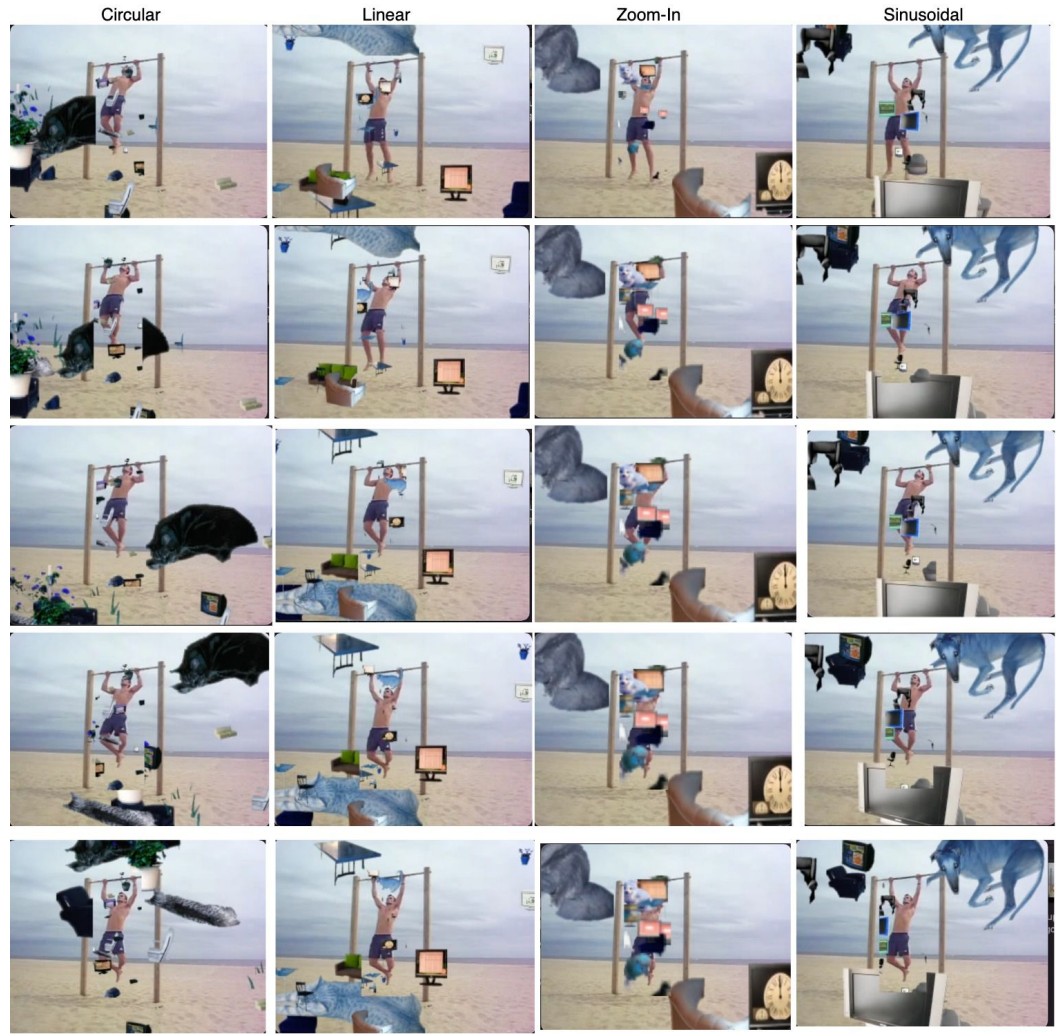

Figure 14: **Different occluder trajectories:** are illustrated for our proposed O-UCF and O-JHMDB datasets.

Indoor Occluders  Outdoor Occluders  Indoor Occluders  Outdoor Occluders

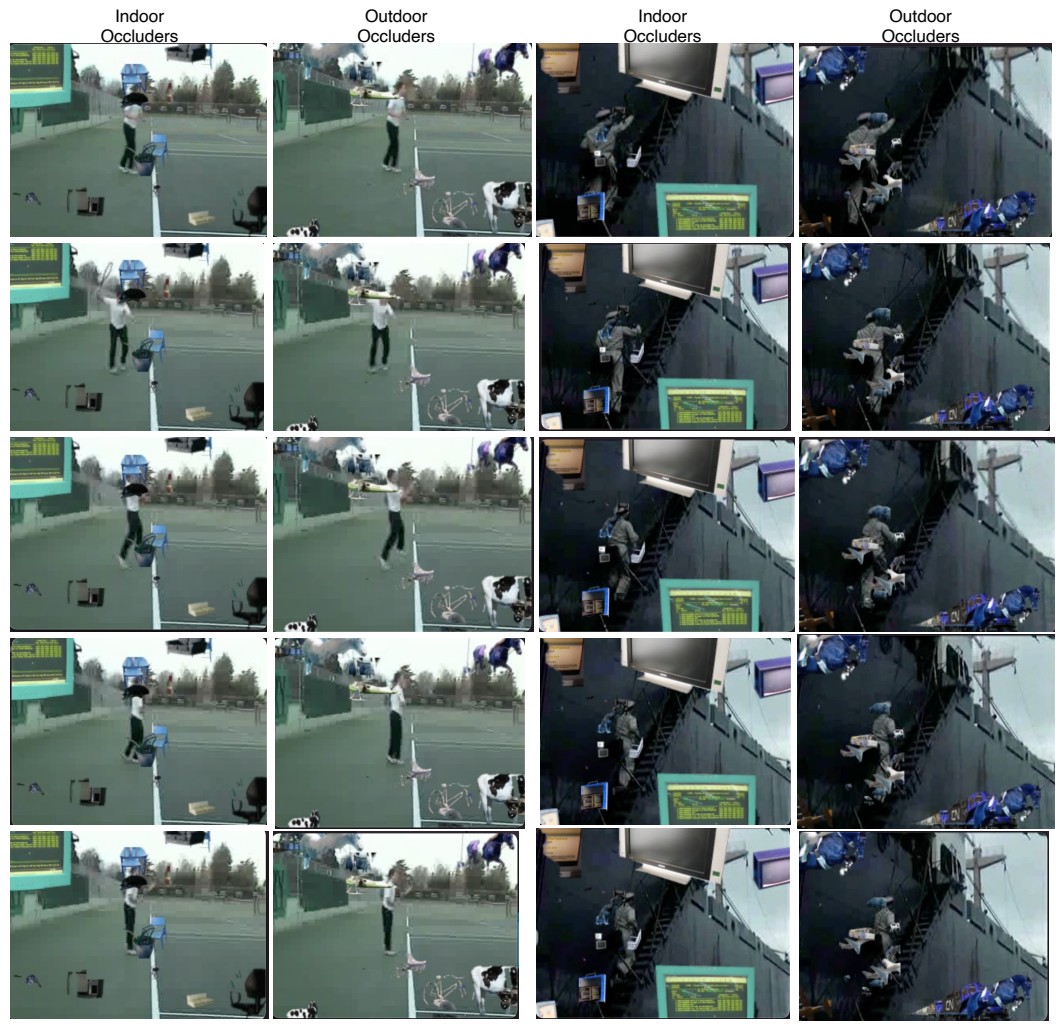

Figure 15: **Nature of Occluders**: Occluders in the proposed O-UCF & O-JHMDB datasets belong to either indoor/outdoor samples.

## A10.    Datasheets For Datasets

### Motivation

**For what purpose was the dataset created?** Was there a specific task in mind? Was there a specific gap that needed to be filled? Please provide a description.

O-UCF, O-JHMDB were created to perform systematic benchmark study of occlusions in video action detection. Real-OUCF was created to evaluate sota action-detectors on real-world occlusions.

**Who created this dataset (e.g., which team, research group) and on behalf of which entity (e.g., company, institution, organization)?**

These were created by a research-group whose identity shall be released after the review process.

### Composition

**What do the instances that comprise the dataset represent (e.g., documents, photos, people, countries)?** Are there multiple types of instances (e.g., movies, users, and ratings; people and interactions between them; nodes and edges)? Please provide a description.

Dataset contains videos of several people performing actions.

**Is there a label or target associated with each instance?** If so, please provide a description.

For each instance, we provide an instance-level segmentation mask along with our datasets.

**Are there recommended data splits (e.g., training, development/validation, testing)?** If so, please provide a description of these splits, explaining the rationale behind them.

The idea for a real-world occlusion dataset is that it is easy to train an existing network using synthetic occlusions as a data augmentation. However, there is a need of consistent real-world tet set to benchmark all the future approaches against, and that is the test set which our Real-OUCF provides.

**Are there any errors, sources of noise, or redundancies in the dataset?** If so, please provide a description.

A very detailed care has been taken to make our annotations noise free. In the rare case that a correction is needed, the updated annotations will be posted on our official link (Shared on top of this manuscript).

**Is the dataset self-contained, or does it link to or otherwise rely on external resources (e.g., websites, tweets, other datasets)?** If it links to or relies on external resources, a) are there guarantees that they will exist, and remain constant, over time; b) are there official archival versions of the complete dataset (i.e., including the external resources as they existed at the time the dataset was created); c) are there any restrictions (e.g., licenses, fees) associated with any of the external resources that might apply to a future user? Please provide descriptions of all external resources and any restrictions associated with them, as well as links or other access points, as appropriate.

All the videos of our dataset were obtained after hand picking from YOUTUBE, and then scraping them.Due to offline availability of the dataset, our dataset is now self-contained.

**Does the dataset relate to people?** If not, you may skip the remaining questions in this section.

Yes, it is spatio-temporal action detection. It primarily considers human actors.

**Does the dataset identify any subpopulations (e.g., by age, gender)?** If so, please describe how these subpopulations are identified and provide a description of their respective distributions within the dataset.

No.

**Is it possible to identify individuals (i.e., one or more natural persons), either directly or indirectly (i.e., in combination with other data) from the dataset?** If so, please describe how.

The classes in our Real-OUCF dataset primarily belong to sports such as basketball. Several videos of the dataset are picked from official sporting events like NBA, Olympics where it might be possible to identify famous athletes by face only. However, all the videos we use are already there in the public domain.

---

### Collection Process

**How was the data associated with each instance acquired?** Was the data directly observable (e.g., raw text, movie ratings), reported by subjects (e.g., survey responses), or indirectly inferred/derived from other data (e.g., part-of-speech tags, model-based guesses for age or language)? If data was reported by subjects or indirectly inferred/derived from other data, was the data validated/verified? If so, please describe how.

Data was raw-video directly observed with human eyes.

**What mechanisms or procedures were used to collect the data (e.g., hardware apparatus or sensor, manual human curation, software program, software API)?** How were these mechanisms or procedures validated?

Manual human curation

**Who was involved in the data collection process (e.g., students, crowdworkers, contractors) and how were they compensated (e.g., how much were crowdworkers paid)?**

Graduate students, who were graciously supported by research grants with warm thanks.

**Over what timeframe was the data collected? Does this timeframe match the creation timeframe of the data associated with the instances (e.g., recent crawl of old news articles)?** If not, please describe the timeframe in which the data associated with the instances was created.

Dataset was collected over a period of 6 months from Dec22-May23.

---

### Preprocessing/cleaning/labeling

**Was any preprocessing/cleaning/labeling of the data done (e.g., discretization or bucketing, tokenization, part-of-speech tagging, SIFT feature extraction, removal of instances, processing of missing values)?** If so, please provide a description. If not, you may skip the remainder of the questions in this section.

Pre-processing was done by temporally cropping a video into smaller clips which indicate start/end of the action. After auto-labelling spatio-temporal masks via SAM, we manually annotate/refine masks using the CVAT tool.

**Was the "raw" data saved in addition to the preprocessed/cleaned/labeled data (e.g., to support unanticipated future uses)?** If so, please provide a link or other access point to the "raw" data.

Link shall be provided soon for the raw data.

**Is the software used to preprocess/clean/label the instances available?** If so, please provide a link or other access point.

Yes, we use normal Segment Anything and Computer Vision Annotation Tool. https://github.com/facebookresearch/segment-anything https://github.com/opencv/cvat

| Uses |
|------|

**Has the dataset been used for any tasks already?** If so, please provide a description.

Yes, for our benchmark study on the impact of occlusions in spatio-temporal video action detection.

**What (other) tasks could the dataset be used for?**

Amodal mask completion, Occlusion Robustness Testing etc.

| Distribution |
|------|

**Will the dataset be distributed to third parties outside of the entity (e.g., company, institution, organization) on behalf of which the dataset was created?** If so, please provide a description.

**How will the dataset will be distributed (e.g., tarball on website, API, GitHub)** Does the dataset have a digital object identifier (DOI)?

**When will the dataset be distributed?**

Just before the start of the neurips 2023 conference.

**Will the dataset be distributed under a copyright or other intellectual property (IP) license, and/or under applicable terms of use (ToU)?** If so, please describe this license and/or ToU, and provide a link or other access point to, or otherwise reproduce, any relevant licensing terms or ToU, as well as any fees associated with these restrictions.

No, its free for use by all parties. However, the discretion to update more video samples in future, and refine the existing annotations over time lie with the original dataset authors.

**Have any third parties imposed IP-based or other restrictions on the data associated with the instances?** If so, please describe these restrictions, and provide a link or other access point to, or otherwise reproduce, any relevant licensing terms, as well as any fees associated with these restrictions.

No, its free for use by all parties. However, the discretion to update more video samples in future, and refine the existing annotations over time lie with the original dataset authors.

**Do any export controls or other regulatory restrictions apply to the dataset or to individual instances?** If so, please describe these restrictions, and provide a link or other access point to, or otherwise reproduce, any supporting documentation.

No, its free for use by all parties. However, the discretion to update more video samples in future, and refine the existing annotations over time lie with the original dataset authors.

| Maintenance |
|------|

**Who will be supporting/hosting/maintaining the dataset?**

Dataset will be supported by the authors/research group of this paper.

**How can the owner/curator/manager of the dataset be contacted (e.g., email address)?**

By email-address, the details shall be revealed post-review .

**Will the dataset be updated (e.g., to correct labeling errors, add new instances, delete instances)?** If so, please describe how often, by whom, and how updates will be communicated to users (e.g., mailing list, GitHub)?

Changes if any , will be posted once a year to our github repo.

**If the dataset relates to people, are there applicable limits on the retention of the data associated with the instances (e.g., were individuals in question told that their data would be retained for a fixed period of time and then deleted)?** If so, please describe these limits and explain how they will be enforced.

**Will older versions of the dataset continue to be supported/hosted/maintained?** If so, please describe how. If not, please describe how its obsolescence will be communicated to users.

Yes, this shall be useful to give reliable comparisons on different dataset versions in existing literature.

**If others want to extend/augment/build on/contribute to the dataset, is there a mechanism for them to do so?** If so, please provide a description. Will these contributions be validated/verified? If so, please describe how. If not, why not? Is there a process for communicating/distributing these contributions to other users? If so, please provide a description.

We also release a python script which allows to create on -the-fly semi-realistic occlusions consisting of static occlusions at different severity levels and different motions of the occluder. Anyone could use this to generate infinite occluder variations.

For realistic occlusions, once could always collect more real world samples or expand the count of existing classes. Note that the test set we provide is already significantly larger than traditional action-detection datasets. [61, 33]

**Any other comments?**

We are grateful to you for taking the time to read this datasheet and reviewing this manuscript.

