# Occlusions in Video Action Detection: Benchmark Datasets And Training Recipes (Supplementary Material)

**Rajat Modi**[1]*, **Vibhav Vineet**[2], **Yogesh Singh Rawat**[1]

CRCV, University of Central Florida[1], and Microsoft Research[2]

## 1 Broader Impact Statement

A decade ago[2], [28] showed that successive filters of a convnet could act as general forms of edge, shape and texture detectors. In Fig 6, we now illustrate that higher layers of a neural net can learn to group pixels into objects at a semantic level without any explicit localization supervision/ multi-modal alignment . This shows that we could learn non-parametric object-queries at the highest levels in the architecture (via unsupervised-clustering), instead of directly injecting learnable parameters/proposals into lower decoder-layers[2]. In nature, there can be a large number of objects present in the retinal frame (eg, leaves of a tree). Depending on the granularity of fixation (eg, separation between multiple trees), we might care only about a subset of those classes[21]. However, the objects we don't care about still exist, even though an object query might not bind to them. Therefore, the encoding process in a neural net should *not* prevent separate islands of finer objects (eg, leaves) from getting formed [3].

The representational collapse issue observed in Sec 5 presents a memory-scaling bottleneck. One way to get around it is to consider the behaviour of a self-replicating cellular-automaton [24] like GLOM[6]: unfolding a singular embedding creates dynamic connectionist hardware on-the-fly[10]. This would also allow us to spiritually succeed the computationally-expensive distillation setup, i.e. simply copying a singular embedding lesser number of times on a weaker hardware would allow the student model to exist. A key question that thus remains unanswered for our community is how to compress the entire knowledge of a neural-net into this singular entity (eg, biological seed) and the mechanism behind its unfolding[13](which is an inverse of the protein-folding problem).

## 2 Mortal computation requires self-replicable embeddings

On the other hand if we want to pack similar levels of intelligence in a brain-like interface which consumes less than 25 watts daily, we have to take an alternate biologically-plausible route[8]: intelligence can be encoded in a single cell (embedding) much like how the genetic code of a human gets encoded in the DNA. In nature, multiple copies of a zygote (cell) lead to emergence of an animal's internal organs. Similarly, in our computers multiple copies of this single replicable embedding shall lead to networks whose structure gets discovered 'on-the-fly' according to local hardware constraints.

It has been well established in distillation literature that [9] a higher-parameterized teacher network can teach a lower-parameter student network to obtain similar performance. If all the knowledge of the teacher could be compressed into a singular embedding, then we won't need distillation at all. Simply copying the single embedding many times on a weaker student hardware would be enough to allow the student model to exist. Although the student model would be weak due to lesser copying,

---

*Corresponding Author, email: rajatmodi@ucf.edu

[2]This section is meant to be philosophical and optimistic in nature.

[3]And instead encode the complete part-whole hierarchy of a scene[4]. This shall resolve the issue of oversegmentation/ inability to distinguish between part-wholes that SAM[16] still faces when given a grid of input point prompts at a very-fine granularity.[16]

the replication step for both student/teacher stems from a common learnt singular representation[4]. These 'connectionist-networks'[10] which start their existence from a single embedding and only remain in the memory as long as the hardware is powered on (hence the name mortal) would require learning algorithms other than backpropogation: where the parameters of each layer could be updated without knowing the precise feed-forward mathematical-functions[7] and allow dynamic growing of the network on a smaller hardware[12]. We remain optimistic for the future that such mortal computation offers to humanity[7].

## 3   Additional Supplementary Material

This manuscript discusses the supplementary materials in addition to our submission. The supplementary material contains seven sections:

- §A1. shows how our simple VCAPS-Mvitv2 model achieves *a new state-of-the-art* in Video Action Detection, *along* with being *robust* to occlusions.
- §A2. proposes a *new task* called action-segmentation involving *instance-level localization* of actor in a video and presents a benchmark to streamline further research in the field.
- §A3. explores the *importance of background context* in action detection.
- §A4. presents the full benchmark and analyzes the background bias property in existing detectors.
- §4 analyzes our results for *synthetic* occluder motions on O-UCF & O-JHMDB datasets.
- §5 discusses more detail about the video-collection and annotation process of our curated Real-OUCF dataset.
- §A1. explores some plausible ways to solve the representational collapse problem in capsules.
- §A2. presents some qualitative samples from the proposed three Benchmark datasets, along with UCF-101 instance-level annotations. All the datasets, benchmarks and codes for this work will be released for free public usage at `https://anonymous.4open.science/r/OccludedActionBenchmark-B9E2`.
- §A3. provides the NeurIPS recommended datasheet explaining the dataset collection mechanism and other important details.

### A1.   A Robust Video-Level State Of The Art

We present a new state-of-the-art in spatio-temporal video action detection specifically on UCF-24 and JHMDB-21 datasets in Tab1. Note that we achieve 83.1% on UCF-24 and 98.1% on JHMDB-21 in terms of the widely accepted[14] v-mAP metric at the 0.5ioU threshold. One of the most desirable properties in an action-detector is that it should perform well on *existing* standard datasets[23] as well as be *robust to occlusions* at the same time. Our simple model namely VCAPS-Mvitv2 achieves the best of both worlds, thereby setting a new video-level state of the art for our community.

The robustness of an action-detector can be measured in two ways, 1) the *actual robustness* under occlusions which has been illustrated as absolute value (i.e. 67.3%) in Tab6 of our original manuscript. 2) Measuring the drop in performance of a detector as the ioU threshold during evaluation is swept from 0.2 to 0.5. We define this quantity as $\kappa = 1 - \left( \frac{vmAP_{0.2} - vmAP_{0.5}}{vmAP_{0.2}} \right)$ which simply measures the relative performance drop from 0.2→0.5ioU. In Tab1, we note that on the much challenging UCF-24 dataset, our method obtains $\kappa = 0.84$, which is greater than all the other methods, thereby indicating more localization robustness. On JHMDB-21, we obtain 92.8% in terms of the absolute v-mAP score, which is significantly better than other existing methods. We acknowledge that TubeR[29] and ST-Mixer [26] are slightly better than our method in terms of f-mAP scores on UCF-24 dataset, although our method is considerably *more* robust (0.84 vs 0.71 on $\kappa$ score).

### A2.   A New Instance Level Benchmark

Traditionally[20, 29], spatio-temporal action-detection has relied on predicting *bounding boxes* across an actor for *each* frame. A much harder task instead would be a *finer-grained* localization, i.e.

---

[4]An undeniable fact of nature is that intelligence/consciousness in humans emerges from self-replication of a singular cell (zygote). It still remains to be seen whether singular prokaryotic organisms like amoeba themselves are conscious [22, 19]

Table 1: **Comparison with existing methods:** Comparison of our method across existing supervised approaches, *: denotes results using a CSN152 backbone. $\kappa = 1 - \left( \frac{vmAP_{0.2} - vmAP_{0.5}}{vmAP_{0.2}} \right)$. Higher value of $\kappa$ denotes more robustness.
.

| | Backbone | | UCF-24 | | | | JHMDB-21 | | | |
| --- | --- | --- | --- | --- | --- | --- | --- | --- | --- | --- |

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

## A1.    Representational Collapse in Capsules

In Fig 5 of the main manuscript, we have shown the problem of representational collapse[6], where capsules cannot decode multiple entities (objects), if the number of objects in the scene are greater than the parameters in the network. A classical way to solve this problem would be to to increase the number of parameters by reinitializing and retraining the machine again. It agrees with the evolutionary observation that more parameters mean more intelligence and the only option to do better is to scale up (i.e 85 billion neurons in the brain vs 100 trillion in ChatGPT4). This is a promising direction if we want to continue to pay exorbitant amounts of money for specialist hardware, expensive electricity or wait for the hardware to become cheaper in future.