# OpenReview forum: "On Occlusions in Video Action Detection: Benchmark Datasets And Training Recipes"
_NeurIPS.cc/2023/Track/Datasets_and_Benchmarks — NeurIPS 2023 Datasets and Benchmarks Poster_

### Official Review · Reviewer_yxN6 · 2023-07-06
**Comments on paper 385**

**Rating:** 6
**Confidence:** 3
**Correctness:** It seems correct.

**Strengths:**

Three datasets are proposed for evaluating video action detection methods in the occlusion situations.

The impact of different aspects are evaluated, including backbone (2D vs 3D, CNN vs transformer), type of occlusion (static vs dynamic), the use of capsule and the use of occlusion as data augmentation. Based on these analyses, an improved training recipe is proposed.



**Additional Feedback:**

See comments above.

**Clarity:**

Not very clear. I am a bit confused about the construction of the Real-OUCF dataset. It would be better to include some examples from the three datasets.

**Documentation:**

Yes.

**Ethics:**

No.

**Limitations:**

yes.

**Opportunities For Improvement:**

I am a bit confused about the difference between the two datasets, O-UCF and Real-OUCF. Are the occlusions synthesized in the same way for the two datasets?

For Tables 5 and 6, the results on Real-OUCF also need to be included. This can reveal how different variants behave in the realistic occlusion situations.

It would be better to include some synthesized examples of the three datasets in the paper.














**Relation To Prior Work:**

It seems so. I did not check carefully.

**Summary And Contributions:**

The paper investigates the effect of occlusions on video action detection. Three specific datasets are constructed: O-UCF, OccJHMDB and RealOccUCF. Several representative video action detection methods are evaluated on the proposed datasets. By analyzing the effects of backbone, data-augmentation and capsule module, a improved training recipe is proposed to better address occlusion in video action detection.

---

> ### Author Response · Authors · 2023-08-22
> **Authors Response to Reviewer yxN6 (1/1)**
>
> We thank the reviewer for their comments and time. Please find our responses as follows:
>
> >> I am a bit confused about the difference between the two datasets, O-UCF and Real-OUCF. Are the occlusions synthesized in the same way for the two datasets?
>
> We apologize for the confusion due to the lack of details on our part. O-UCF is a synthetic occlusion dataset. Here occluders are chosen from an existing dataset PASCAL, and just pasted onto the frame as a type of alpha-matting. For Real-OUCF, we don't synthesize any occlusions, the occlusions are purely natural. For eg, imagine two people doing a salsa -dance. Both of them touch each other partially, so this is a case of partial occlusion. Another example is a person walking on the street and suddenly a tree/car blocking a part of the person from the camera view. These are occlusions which occur in the real-world and we have tried to curate in Real-OUCF. More details on how videos were selected have been now mentioned in lines 93-112 in the paper. We have also added an extra section to the supplementary.
>
>
> >> For Tables 5 and 6, the results on Real-OUCF also need to be included. This can reveal how different variants behave in the realistic occlusion situations.
>
>
> We now include the results on Real-OUCF in tables 5 and 6. In table 5, it can be seen that models trained with occlusions as augmentation perform better than models without occlusions as augmentation. Similarly, in Table 6, Mvitv2 (transformer), seems to be exhibiting more robustness as compared to the other backbones.
>
> >>It would be better to include some synthesized examples of the three datasets in the paper.
>
> We thank the reviewer for their interest in the dataset. Some of the samples from the dataset are shared [here](https://anonymous.4open.science/r/OccludedActionBenchmark-B9E2/rebuttal/datasets/). We have also updated the figure 2 of the main manuscript, and provided additional samples in the figures 3/4 of the supplementary. We will also release all the datasets before the start of the conference, along with further-cleaned annotations.
>
> Thank you so much for all the comments and we hope that we were able to answer some of the questions. Looking forward to engage in further discussions.
>
> Yours sincerely,
> Paper 385 authors

---

> > ### Author Response · Authors · 2023-08-29
> > **Request for a chance of any further clarifications**
> >
> > Respected Reviewer yxN6:
> >
> > We hope you are doing well, We were reaching out in case there were further clarifications we could provide in regards to our work. We hope that we were able to address the concerns raised by the reviewer. In case of any further issues, we will be grateful for a chance to provide further resolutions.
> >
> > We thank you for the valuable time and effort to review our submission,
> >
> > Yours sincerely,
> >
> > Authors of Paper 385

---

### Official Review · Reviewer_2htk · 2023-07-21
**A good paper exploring occlussions for the spatial-temporal action detection task that needs some polish**

**Rating:** 7
**Confidence:** 4

**Strengths:**

The authors claim it is the first video occlusion dataset that focuses on the localization ability of SOTA spatial-temporal action detectors. Interestingly, Mvitv2 is robust to occlusions even when not exposed to them. It is a relevant pursuit considering the issues in self-driving cars.

**Additional Feedback:**

Although it is appreciated that there are nine severity levels, the reasons behind how the movement of occluders, occluder distribution considerations and what makes the occluder realistic could be further bolstered.

**Clarity:**

Aside from the language and grammar issues, the paper is mostly clear with accessible language.

**Correctness:**

Most claims made in the paper are appropriate, with relevant accepted metrics reported.

**Documentation:**

The formulation of the occlusions in terms of composition and severity is clear, but I would have liked to receive access to the samples and the annotations created to vet the quality of the samples.

**Ethics:**

Assuming the original datasets (UCF24 and JHMDB) were ethically corrected and the usage agreements are adhered to, the derivative works in this article, there are no ethical concerns.

**Limitations:**

The authors are not as forthcoming about context-specific limitations of the work and only pose limitations relevant to the community at large (lines 264-269).

**Opportunities For Improvement:**

The paper could do with a language and grammar check to improve the work's readability. There could also be more experimentation related to unpacking why Mvitv2 performs the way it does.

**Relation To Prior Work:**

Although there are adjacent works that work on occlusions for action recognition and video instance segmentation, the datasets and insights gained are novel.

**Summary And Contributions:**

The authors propose three derivative datasets that benchmark the impact of occlusions for the spatial-temporal action detection task. The paper explicitly covers variations in 2D and 3D backbones, a comparison occlusion tolerance for transformer (Mvitv2) and varying CNN backbones (DLA34, ResNext, i3D), along with recommendations such as leveraging capsules and masking input tokens to achieve better occlusion robustness.

---

> ### Author Response · Authors · 2023-08-22
> **Authors Response to Reviewer 2htk (1/2)**
>
> We thank the reviewer for their very helpful suggestions and feedback. Please find our responses below:
>
> >>The paper could do with a language and grammar check to improve the work's readability. There could also be more experimentation related to unpacking why Mvitv2 performs the way it does.
>
> We have corrected the spelling mistakes in the revised manuscript now.
>
> For unpacking why MvitV2 performs the way it does, we have added an additional section 5.3, lines 256- 270 and new qualitative results  (Fig 7 ) now. As can be now seen in the figure, the obtained semantic maps show clear evidence of the transformer forming clusters of objects in the scene (i.e. actor/occluders). Next, the later capsule layers are selectively binding to one of the objects. Finally, the decoder is learning to suppress occluder specific features and reliably localize actors.
>
> Such semantic maps had been called islands of agreement by Hinton Et Al[1], but remained purely hypothetical till early 2022. In last year CVPR'22 one work[2] did demonstrate such islands emerging for datasets like MNIST. Now, we are excited to see a similar pattern emerge in *realistic* datasets like JHMDB, and show the general nature of representations being learnt by transformers like Mvitv2.
>
> We have shared such semantic maps obtained on the O-JHMDB dataset [here](https://anonymous.4open.science/r/OccludedActionBenchmark-B9E2/rebuttal/semantic_islands/v)
>
> >>The authors are not as forthcoming about context-specific limitations of the work and only pose limitations relevant to the community at large (lines 264-269).
>
> Thank you so much for motivating us to be more transparent. We have now rewritten the limitations sections, to highlight context-specific limitations.
>
> *In this work, we have focused on annotating visible regions of the occluded-objects in line with official COCO protocol[34]. An interesting direction could also focus on predicting missing regions, i.e. amodal-segmentation. Classically, semantic-segmentation has assumed that one pixel can belong to only one object. However, this is not true for occlusions. Inductively, this symmetry issue has been resolved in works like Maskformer2 where multiple objects per pixel can be predicted by removing the softmax assumption. This will be helpful in solving the occlusion problem and at the same time help overcome the perceptual-crowding assumption made in capsules. We could also have explored if explicit SSL-reconstruction objectives like MIM could improve downstream occlusion robustness. Finally, we note that there is still a significant gap left to bridge in existing models for improving robustness to realistic occlusions.*
>
> >> I would have liked to receive access to the samples and the annotations created to vet the quality of the samples.
>
> We thank the reviewer for their interest in the proposed dataset, We provide some samples [here](https://anonymous.4open.science/r/OccludedActionBenchmark-B9E2/rebuttal/datasets/), with overlayed semantic-annotations for convenience. The dataset at full resolution is more than 100 Gigs in size, so we are still in the process of figuring out the hosting. We will release the dataset (along with a second round of cleaned annotations) before the start of the conference.
>
> >>Although it is appreciated that there are nine severity levels, the reasons behind how the movement of occluders, occluder distribution considerations and what makes the occluder realistic could be further bolstered.
>
> We apologize for the lack of clarity and will be grateful for a chance to provide clarifications. Movements of the occluder could only be made realistic if we knew in advance the precise trajectory of the occluder, along with instance-level mask. That information is present in the OVIS dataset, and we have incorporated those experiments in the manuscript now. For realistic occlusion, there are two cases we define: 1) semi-real: consider for example an occluder from OVIS superimposed on a UCF frame. Here, the trajectory of the occcluder will be realistic. However, when we actually place the occluder on the frame, there is  a disconnect between the actual occluder and the frame. Therefore, it cannot be truly realistic. Hence, we resort to the term semi-real to highlight that it has realistic motion. 2) real: consider for example an object, occluded naturally by a tree. Or, two humans doing salsa-spin together. They mutually occlude each other. If a camera recorded them dancing, the entire captured occlusion is natural, with no manual modifications.  Such natural occlusions are present in our dataset Real-OUCF. Therefore, we have tried to experiment with all the variations, i.e. synthetic, semi-real and real in this work.
>
> We thank the respected reviewer for their valuable comments and positive feedback. We look forward for a chance to engage in further discussions.
>
> Yours sincerely,
>
> Paper 385 authors
>
> P.S. References are mentioned in the next comment.

---

> > ### Author Response · Authors · 2023-08-22
> > **Authors Response to Reviewer 2htk (2/2)**
> >
> > References
> >
> > [1] Hinton, Geoffrey. "How to represent part-whole hierarchies in a neural network." Neural Computation (2022): 1-40.
> >
> > [2] Garau, Nicola, et al. "Interpretable part-whole hierarchies and conceptual-semantic relationships in neural networks." Proceedings of the IEEE/CVF Conference on Computer Vision and Pattern Recognition

---

> > > ### Author Response · Authors · 2023-08-29
> > > **Request for a chance of any further clarifications**
> > >
> > > Respected Reviewer 2htk:
> > >
> > > We hope you are doing well, We were reaching out in case there were further clarifications we could provide in regards to our work. We hope that we were able to address the concerns raised by the reviewer. In case of any further issues, we will be grateful for a chance to provide further resolutions.
> > >
> > > We thank you for the valuable time and effort to review our submission,
> > >
> > > Yours sincerely,
> > >
> > > Authors of Paper 385

---

### Official Review · Reviewer_NCed · 2023-07-26
**Good idea, useful study, messy execution**

**Rating:** 6
**Confidence:** 4

**Strengths:**

- Interesting study about impact of occlusion in spatiotemporal action localization.
- Useful datasets / benchmarks.
- Good experimental breadth with several models analyzed.

**Additional Feedback:**

Main issue is clarity of paper, which is hard to follow.

--- raised rating post-rebuttal as the authors improved the paper by using dynamic occluders from a video segmentation dataset.

**Clarity:**

This is the main problem of the paper -- the paper is hard to follow, many things are more complicated than they should be, the writing is not very consistent (e..g the abstract refers to the same dataset as RealOccUCF and Real-OUCF).

Maybe some of the experiments should be moved to appendix, then the main ones would be explained in more detail in the main paper.

**Correctness:**

Seems correct, but too little details about some of the data to evaluate --- namely the RealOccUCF dataset.

**Documentation:**

Should add more detail about Real dataset.

**Ethics:**

No concerns.

**Limitations:**

Limitations addressed.

**Opportunities For Improvement:**

The writeup is very hard to follow:
- For example the real dataset, it's not clear what the goal of the annotations is. I suppose that some of people are occluded, and somehow this information is derived from the collected segmentations. What fraction of people annotated are occluded ?
- Some things are more complicated that they have to be -- for example tables 2 and 3 mention "Performance across 9 occlusion levels", but only 3 are shown, then it's implied that 3 of them are averaged over.
- Why not use directly occluders from video, like from OVIS, instead of PASCAL VOC, which then requires synthesising artificial (simplistic) motions in the dynamic case.
- There are too many results with too little detail, some of them seem superfluous -- table 4 has too many columns, and seems like static vs dynamic occluders doesnt make a big difference.

**Relation To Prior Work:**

Discussed.

**Summary And Contributions:**

The paper introduces datasets for evaluating the impact of occlusion in spatiotemporal action localization.
It gets segmented objects from the PASCAL VOC dataset and pastes them on top of videos (static occluders), potentially with simple motion (dynamic occluders) as well as non-synthetic custom dataset collected from youtube and annotated semi-automatically with segmentation. The paper then goes to compare the performance of different spatiotemporal action localization models on the three datasets and draws conclusions.

---

> ### Author Response · Authors · 2023-08-22
> **Authors Response to Reviewer NCed (1/1)**
>
> We thank the reviewer for their valuable time, helpful suggestions, and questions. We are grateful to provide our responses as follows:
>
> >>For example the real dataset, it's not clear what the goal of the annotations is. I suppose that some of people are occluded, and somehow this information is derived from the collected segmentations. What fraction of people annotated are occluded ?
>
> 64.1% of the people are occluded in the Real-OUCF dataset. Indeed,we arrived at this number using the intersection of the collected segmentations. We have updated line 129 with this figure now.
>
> >>Some things are more complicated that they have to be -- for example tables 2 and 3 mention "Performance across 9 occlusion levels", but only 3 are shown, then it's implied that 3 of them are averaged over.
>
> We apologize for the mistake on our part. We have updated tables 2 and 3 to reflect that the reported results are averaged over 3 background severity levels for a particular foreground level. The full benchmark with all 9 severity levels has been mentioned in Tables 4 and 5 of the supplementary, along with an additional section on background bias analysis.
> >>Why not use directly occluders from video, like from OVIS, instead of PASCAL VOC, which then requires synthesising artificial (simplistic) motions in the dynamic case.
>
> Thank you so much for this valuable insight, earlier we were also surprised that our experiments on controlled dynamic motions did not yield significant differences with static cases. Following your very helpful advice, we were able to analyze the behaviour on OVIS occluders. Specifically, we pick occluders from OVIS, and superimpose on UCF/JHMDB datasets. The motions of OVIS occluders can be considered realistic. We generate two sets: a) where the  occluders are static, b) where the occluders are moving. Some of our created samples are available [here.](https://anonymous.4open.science/r/OccludedActionBenchmark-B9E2)
>
> Following are the results:
> |       | using    | original | backbones  |          |
> |-------|----------|----------|------------|----------|
> |       | Ovis-UCF |          | Ovis-JHMDB |          |
> |       | Static   | Dynamic  | Static     | Dynamic  |
> | MOC   | **14.8** | 12.4     | 26.8       | **31.9** |
> | YOWO  | **16.5** | 10.1     | **37.6**   | 35.3     |
> | VCAPS | **32.3** | 21.7     | **20.9**   | 17.3     |
> |       |          |          |            |          |
> |       | using    | resnet18 | backbones  |          |
> |       | Ovis-UCF |          | Ovis-JHMDB |          |
> |       | Static   | Dynamic  | Static     | Dynamic  |
> | YOWO  | **17.3** | 11.8     | **28.0**   | 24.2     |
> | VCAPS | **33.2** | 21.9     | **7.7**    | 7.3      |
>
> As we can see, in general models are performing better during static occlusions than dynamic occlusions. The differences still hold when the backbones are  made consistent. Therefore, we conclude that :
> 1) For realistic motions on OVIS based datasets, models perform better for static cases then dynamic. This shows that reasoning about temporal motion properly remains a worthwhile pursuit for our models.
> 2) For controlled motions, the differences are close, and we cannot make any definite conclusion using those.
> We have updated lines 180-186 of the manuscript accordingly.
>
> >>There are too many results with too little detail, some of them seem superfluous -- table 4 has too many columns, and seems like static vs dynamic occluders doesnt make a big difference.
>
> We have now  reduced the number of columns in Table 4 to only compare between static and dynamic occluders. Thank you so much for helping us improve clarity.
>
> >>but too little details about some of the data to evaluate --- namely the RealOccUCF dataset.
>
> We have added more details to the table 8b). Some more details about how the Real-OUCF dataset was collected and annotated are presented in Section 6 of the revised supplementary. We have also shared some of the samples of the dataset [here](https://anonymous.4open.science/r/OccludedActionBenchmark-B9E2/rebuttal/datasets/realoucf/readme.md)
>
> >>This is the main problem of the paper -- the paper is hard to follow, many things are more complicated than they should be, the writing is not very consistent (e..g the abstract refers to the same dataset as RealOccUCF and Real-OUCF). Maybe some of the experiments should be moved to appendix, then the main ones would be explained in more detail in the main paper.
>
> We have reworded RealOccUCF to Real-OUCF. We apologize for the confusion.
>
> We have moved the experiments with occluder motions (table 4) to suppllementary now based on your valuable suggestion. Instead, we now discuss the results of static vs dynamic occlusions on OVIS datasets in the lines 180-187 of the main manuscript.
>
> We thank the reviewer for their very helpful suggestions on the constructive experiments we were able to perform. It shall mean a lot to us to be able to answer any further questions.
>
> Yours sincerely,
>
> Paper 385 authors

---

> > ### Author Response · Authors · 2023-08-29
> > **Request for a chance of any further clarifications**
> >
> > Respected Reviewer NCed:
> >
> > We hope you are doing well, We were reaching out in case there were further clarifications we could provide in regards to our work. We hope that we were able to address the concerns raised by the reviewer. In case of any further issues, we will be grateful for a chance to provide further resolutions.
> >
> > We thank you for the valuable time and effort to review our submission,
> >
> > Yours sincerely,
> >
> > Authors of Paper 385

---

> > ### Comment · Reviewer_NCed · 2023-08-29
> > **good effort**
> >
> > That's a good upgrade on the paper with OVIS. I think the paper may be spread too thin going into too many directions and could benefit from some more focus, but overall i'm happy to increase my rating.

---

> > > ### Author Response · Authors · 2023-08-29
> > >
> > > Respected Reviewer NCed:
> > >
> > > We are grateful for the kind consideration you have given to our work. The experiments with OVIS helped us better understand the dynamic aspects of occluders and we hope that our efforts were able to better explain those aspects in the revised draft. We will try to further improve the focus of the paper and improve the engagement of a prospective reader.
> > >
> > > Thank you so much,
> > >
> > > Yours sincerely,
> > >
> > > Paper 385 Authors.

---

### Official Review · Reviewer_cTJi · 2023-07-28
**Review for "On Occlusions in Video Action Detection: Benchmark Datasets And Training Recipes"**

**Rating:** 3
**Confidence:** 4

**Strengths:**

1. This paper presents the first systematic analysis of the detrimental effects of occlusion and proposes corresponding solutions.

2. The visual analysis of the capsule module is intriguing and provides valuable insights.

**Additional Feedback:**

1. The significant performance variations among the three methods on different datasets, as shown in Table 2 and Table 3, require further analysis from the authors. Providing additional insights into the factors influencing these differences would be beneficial.

2. What if using $\delta_a$ instead of $\delta_r$ in Figure 3, 4?

**Clarity:**

Concerning the dataset construction, the motivation is clearly stated. However, there are some issues with the comparison subjects and the interpretation of the table components.

In addition, a few minor issues are observed:

1. L179: There is a broken hyperlink.

2. The titles "Are backbones with more parameters necessarily more robust?" and "Increasing parameter size improves occlusion robustness" appear somewhat contradictory. The latter title could be rephrased to avoid this conflict.

**Correctness:**


My major concerns are as follows:

1. Many of the conclusions proposed by the authors lack strong experimental support.

2. The synthetic datasets constructed may significantly differ from real-world scenarios. Thus, the conclusions drawn from the synthetic datasets might not directly apply to real-world situations. Moreover, the experimental results provided on the RealOccUCF dataset appear to be limited.

**Documentation:**

It seems neither code nor dataset is available in current submission.

**Ethics:**

No.

**Limitations:**

The authors have adequately addressed the limitations concerning the dataset in their paper.

**Opportunities For Improvement:**

1. The conclusion drawn in Section 4.1, attributing all differences in occluder motion analysis to variations in the backbone, seems unreasonable. Since the analyzed methods differ in their overall frameworks and use different loss functions, it is essential to control for variables properly before inferring which specific aspect contributes to the observed performance differences.

2. The authors did not provide sufficient details about training different backbones for YOWO and VCAPS. Training a transformer-based model from scratch requires longer training times compared to pre-training a CNN network. The current experimental results do not conclusively demonstrate that pre-training has the most significant impact.

3. The exploration of parameter size only focuses on the ResNet series, which is insufficient to support the conclusion. It would be more comprehensive to include comparisons with different models like MVITv2 and ResNeXt.

4. Table 8(b) lacks explicit clarification on which models utilized Token masking. Moreover, the four methods in this table have different structures and backbones, making it difficult to control for variables properly. Additionally, the improvements shown in Table 7 appear to be limited.

**Relation To Prior Work:**

Yes.

**Summary And Contributions:**

This paper presents a meticulous analysis of occlusion's influence on video action detection, using three datasets: O-UCF, OccJHMDB (synthetic datasets), and RealOccUCF (realistic dataset). Specifically, the authors investigate: 1) how occlusion types and severity affects the performance; 2) the robustness of three classical methods; 3) the effect of training strategy. At last, the authors exploit data augmentation strategy to improve models' robustness.

---

> ### Author Response · Authors · 2023-08-22
> **Authors Response to Reviewer cTJi (1/3)**
>
> We thank the reviewer for their valuable comments and detailed questions. Please refer to our responses below:
>
> >> The conclusion drawn in Section 4.1, attributing all differences in occluder motion analysis to variations in the backbone, seems unreasonable. Since the analyzed methods differ in their overall frameworks and use different loss functions, it is essential to control for variables properly before inferring which specific aspect contributes to the observed performance differences.
>
> A) Thank you so much for pointing this out. We agree that the differences in the static vs dynamic cases for *controlled* trajectories were marginal for making stronger conclusions. Therefore based on the recommendation of reviewer NCed, we have performed additional experiments with *realistic* occluder motions on the OVIS dataset, and present those results in table 4 of the revised manuscript.
>
> Accordingly, we have revised our wording in line 182: "in general models are performing better during static occlusions than dynamic occlusions". For a more detailed response, we refer to our reply to reviewer NCed.
>
> **Consistent frameworks (losses and backbones) for benchmarking**: In this work, we have studied three methods namely MOC,YOWO and VideoCapsuleNet using hyperparameter settings of the respective papers which were used to obtain SOTA results in their works. While there are several confounding variables (eg, losses, backbones) present, however, consistifying these variables would result in breaking the experimental setup as follows:
>
> i)Different Problem Formulations lead to different losses:  Methods like VideoCapsuleNet pose action detection as a semantic per-pixel classification problem. Therefore, they use a Dice loss. On the other hand, methods like MOC explicitly only regress 4 coordinates of a bounding box (based on center-net formulation) and use well established ioU based losses. Therefore, the same problem has been approached in different ways (classification vs regression) which leads to usage of different loss terms.
>
> ii) Different backbones: While MOC consists of a 2D backbone, other methods consist of a 3D backbone. To the best of our knowledge, we are not aware of any action detection methods in which 2D and 3D backbones can be interchangebly swapped.
>
> Thus, any experiment we perform will have some confounding variables present; all we can do is try and reduce the impact of these other variables. Towards that end, in Table 5 we had shown the results with consistent backbones for YOWO/VideoCapsuleNet, which *both* used a 3D backbone and could be reliably made consistent. Finally, we note that existing benchmarking studies have also used original backbones of the methods being benchmarked[1].
>
> >>The authors did not provide sufficient details about training different backbones for YOWO and VCAPS. Training a transformer-based model from scratch requires longer training times compared to pre-training a CNN network. The current experimental results do not conclusively demonstrate that pre-training has the most significant impact.
>
> B) Thank you so much for the deep insight. We apologize for having missed mentioning the appropriate details. Indeed, the transformer based models were trained for a 1.5X longer schedule due to delayed convergence. We have mentioned the same in lines 135-137 of the revised manuscript. The peak performance (i.e. 24.2) of transformers trained w/o weights for eg, VCAPS-Mvitv2 (on O-UCF)  was reached on epoch 44, whereas we had trained the models for 60 epochs.
>
> Regarding the effects of pretraining, figure 3 presents the results of 3 methods across 2 methods for training both w and w/o pretrained weights. We believe that consistent trend across multiple methods and datasets is a strong indicator that pre-training helps improve natural occlusion robustness of models. Accordingly, we have removed the word "largely" and mentioned in lines 167,172 that pre-training seems to help improve intrinsic robustness.

---

> > ### Author Response · Authors · 2023-08-22
> > **Authors Response to Reviewer cTJi (2/3)**
> >
> > >>The exploration of parameter size only focuses on the ResNet series, which is insufficient to support the conclusion. It would be more comprehensive to include comparisons with different models like MVITv2 and ResNeXt.
> >
> > |          | Parameters | Clean | Occ  |
> > |----------|------------|-------|------|
> > | MViTv2-S | 34.5       | 83.1  | 67.3 |
> > | MViTv2-B | 51.2       | 84.5  | 68.2 |
> > | MViTv2-L | 217.6      | -     | -    |
> >
> > C) As we can see Mvitv2-B shows better performance on both clean and occluded data. For MViTv2-L, the model is taking a lot of time to train owing to almost 5x parameters of MViTv2-B. We will update the results here as soon as our model finishes training.
> >
> > For ResNext family, YOWO has originally used ResNext101 with pre-trained Kinetics weights. However, for ResNext 50/152, official Kinetics weights are not available[2]. Therefore, we were not able to perform those experiments. While official torchvision does provide pre-trained imagenet weights, for our task, i.e action-detection , Kinetics weights offer a well-accepted prior in practice[3].
> >
> > >>Table 8(b) lacks explicit clarification on which models utilized Token masking. Moreover, the four methods in this table have different structures and backbones, making it difficult to control for variables properly. Additionally, the improvements shown in Table 7 appear to be limited.
> >
> > D) Thank you so much for the deep observation. We have revised table 8b) to make it more clear. a) MOC, VCAPS, and YOWO in the table are models which use occlusion as augmentation + dropout in intermediate neurons during training. b) VCAPS-Mvitv2 uses occlusion as augmentation + token masking during training. Since MOC, VCAPS, and YOWO are originally CNN based methods, token-based masking was not possible in them. Therefore, we had experimented with next-nearest inductive bias, i.e. dropping intermediate neurons. We have shown the results for the same on Table 7 also. We have added more clarifications in the tables in the revised version now.
> >
> > For consistent experimental setup, we hope our previous comment A) is able to provide some insight on the chosen experimental settings.
> >
> > Thank you so much for pointing out the limited performance gains in Table 7. We note that while token masking does perform better than all other methods, there is a significant margin of improvement left to bridgle. For eg, on Real-OUCF the best performance is 14.3%. This shows that occlusions are still an open-problem and remains a worthwhile pursuit. We have noted this as a limitation in lines 291-292 of the revised draft.  Finally, we would like to iterate that our VCAPS-Mvitv2 trained with our proposed traiing recipes *surpasses* all the state-of-the-arts in action detection for UCF/JHMDB datasets (Table 1, supplementary), thereby indicating it's effectiveness.
> >
> > >>Many of the conclusions proposed by the authors lack strong experimental support
> >
> > E) We hope that our additional experiments with static vs dynamic motions under realistic settings, more evaluation results on real-oucf (tables 5 and 6), and additional qualitative analysis is able to provide some more clarifications. We have also provided the results with different parameter sizes in Mvitv2 family above.
> >
> > >>L179: There is a broken hyperlink.
> > F) We apologize for the confusion and formatting issues on our end.
> >
> > We have now fixed line 177 hyperlink to point to Fig 4.
> >
> > >>The titles "Are backbones with more parameters necessarily more robust?" and "Increasing parameter size improves occlusion robustness" appear somewhat contradictory. The latter title could be rephrased to avoid this conflict.
> >
> > G) Thank you so much for pointing this out. We have updated line 174 as "Increasing parameter size in the *same* model family improves occlusion robustness".

---

> > > ### Author Response · Authors · 2023-08-22
> > > **Authors Response to Reviewer cTJi (3/3)**
> > >
> > > >>The significant performance variations among the three methods on different datasets, as shown in Table 2 and Table 3, require further analysis from the authors. Providing additional insights into the factors influencing these differences would be beneficial.
> > >
> > > H) We provide additional insights for the difference in the performance among different methods in the lines 151-157 now. The larger performance difference in MOC and YOWO could be attributed to the treatment of action- detection as a box-regression problem in MOC/YOWO vs dense-semantic mask prediction in the VideoCapsuleNet. Regressing the 4 coordinates of a box has a minimum probability of error as 1/4 = 0.25, whereas dense-prediction reduces this error to 1/n, where n is the no of pixels being predicted and n >> 4. As long as at least four pixels are predicted along the boundaries of the box, the method performs relatively well. Also capsules have been shown to be extremely robust to occlusions on Multi-MNIST, and we believe similar inductive bias extends to videos. Furthermore, we show some qualitative justification of why capsules perform well in Fig 5). Finally, for reasons that Mvitv2 works well, we have added an additional section (Fig 7).
> > >
> > > >>The synthetic datasets constructed may significantly differ from real-world scenarios. Thus, the conclusions drawn from the synthetic datasets might not directly apply to real-world situations. Moreover, the experimental results provided on the RealOccUCF dataset appear to be limited.
> > >
> > > We agree that there might be significant differences between synthetic and real world conditions. Therefore, that motivated us to collect the realistic dataset Real-OUCF. Testing extensively on Real-OUCF in the updated manuscript (Tables 5,6,7) shows similar trends as observed on the synthetic data, for eg, occlusion as augmentation improving robustness etc. However, we note that although we surpass all the other methods in Table 8b), there is still a large degree of gap remaining. Thank you so much for pointing this out as a limitation. We have added this as a limitation in lines 291-293 now.
> > >
> > >
> > >
> > > We hope that we are able to answer the reviewers questions, and we will be happy to respond to further clarifications. Thank you so much for giving us your valuable time.
> > >
> > >
> > > Yours sincerely,
> > > Paper 385 authors
> > >
> > > [1] Chantry, Madeline, et al. "Robustness Analysis of Video-Language Models Against Visual and Language Perturbations." Advances in Neural Information Processing Systems 35 (2022): 34405-34420.
> > >
> > > [2] https://github.com/wei-tim/YOWO
> > >
> > > [3] MViTv2: Improved Multiscale Vision Transformers for Classification and Detection

---

> > > > ### Author Response · Authors · 2023-08-29
> > > > **Request for a chance of any further clarifications**
> > > >
> > > > Respected Reviewer cTji:
> > > >
> > > > We hope you are doing well, We were reaching out in case there were further clarifications we could provide in regards to our work. We hope that we were able to address the concerns raised by the reviewer. In case of any further issues, we will be grateful for a chance to provide further resolutions.
> > > >
> > > > We thank you for the valuable time and effort to review our submission,
> > > >
> > > > Yours sincerely,
> > > > Authors of Paper 385

---

### Author Response · Authors · 2023-08-22
**Author's global response to all the respected reviewers**

We thank all the reviewers for their valuable time and insightful comments. We appreciate that the reviewers recognize our work as the first study of occlusions in action detection (Reviewer cTJi, Reviewer 2htk), providing useful datasets/benchmarks with extensive experimental breadth (NCed and yxN6), an intriguing analysis of capsules( Reviewer cTJi) and a relevant pursuit considering the issues in self-driving cars (Reviewer 2htk). To further improve the quality of our submission, we have made the following major changes (including other minor updates) to our work:

1. We present two new datasets called  OVIS-UCF and OVIS-JHMDB for studying how models behave under *realistic* occluders from OVIS dataset. We have revised our claims for *controlled* motions appropriately, owing to close differences in the original table 4.

2.  We now qualitatively unpack the effectiveness of our model VCAPS-Mvitv2 and present the notion that *islands of agreement can emerge from token representations*.

3. We release dataset samples [here](https://anonymous.4open.science/r/OccludedActionBenchmark-B9E2/rebuttal/datasets/) along with exciting visualizations of islands of agreement [1].

We hope that these examples shed some more light on how transformers (eg, Mvitv2) perform so well on occlusion conditions. We have also uploaded the revised manuscript and supplementary with the changes marked in blue.

Yours sincerely,

Paper 385 Authors.

References:

[1] Hinton, Geoffrey. "How to represent part-whole hierarchies in a neural network." Neural Computation (2022): 1-40.

---

### Decision · Program_Chairs · 2023-09-22

**Decision:**

Accept (Poster)

**Comment:**

It seems the authors have tried their best to address the issues raised by the reviewers.  Unfortunately, the first reviewer did not give a second round of comments based on the rebuttal.    The three datasets proposed in the occlusion settings could be interesting for researchers who are doing video action detection.   The authors are encouraged to further improve their work based on the reviews.  This decision has been made in agreement with the AC and SAC.